# Metabolomic machine learning predictor for diagnosis and prognosis of gastric cancer

Yangzi Chen[1,12], Bohong Wang[1,2,12], Yizi Zhao[1,12], Xinxin Shao[3,12], Mingshuo Wang[1,2,12], Fuhai Ma[3,4,12], Laishou Yang[5], Meng Nie [1], Peng Jin[3,6], Ke Yao [1], Haibin Song[7], Shenghan Lou[5], Hang Wang[5], Tianshu Yang[8,9], Yantao Tian [3] ✉, Peng Han [10,11] ✉ & Zeping Hu [1,2] ✉

Gastric cancer (GC) represents a significant burden of cancer-related mortality worldwide, underscoring an urgent need for the development of early detection strategies and precise postoperative interventions. However, the identification of non-invasive biomarkers for early diagnosis and patient risk stratification remains underexplored. Here, we conduct a targeted metabolomics analysis of 702 plasma samples from multi-center participants to elucidate the GC metabolic reprogramming. Our machine learning analysis reveals a 10-metabolite GC diagnostic model, which is validated in an external test set with a sensitivity of 0.905, outperforming conventional methods leveraging cancer protein markers (sensitivity < 0.40). Additionally, our machine learning-derived prognostic model demonstrates superior performance to traditional models utilizing clinical parameters and effectively stratifies patients into different risk groups to guide precision interventions. Collectively, our findings reveal the metabolic landscape of GC and identify two distinct biomarker panels that enable early detection and prognosis prediction respectively, thus facilitating precision medicine in GC.

Gastric cancer (GC) is a highly lethal cancer worldwide[1]. Early diagnosis is critical for improving clinical outcomes by facilitating timely intervention[2]. However, the gold standard for diagnosing GC, which is the endoscopic examination, is both invasive and costly[3–5], limiting its clinical application. Consequently, there is an urgent need for non-invasive detection approaches with high sensitivity and specificity. Additionally, prompt disease management through prognostic surveillance contributes to better clinical outcomes[6]. Currently, clinical prognosis prediction relies heavily on surgeons' empirical judgment based on various clinical indications, including tumor location, TNM staging information, and histopathology, which exhibit limited accuracy[7–9]. Therefore, developing a more precise method for predicting patients' outcomes and stratifying them into different risk groups for appropriate interventions is crucial.

[1]School of Pharmaceutical Sciences, Tsinghua University, Beijing 100084, China. [2]Tsinghua-Peking Joint Center for Life Sciences, Tsinghua University, Beijing 100084, China. [3]National Cancer Center, National Clinical Research Center for Cancer, Cancer Hospital, Chinese Academy of Medical Sciences, Peking Union Medical College, Beijing 100730, China. [4]Department of General Surgery, Department of Gastrointestinal Surgery, Beijing Hospital, National Center of Gerontology, Institute of Geriatric Medicine, Chinese Academy of Medical Sciences, Beijing 100730, China. [5]Department of Colorectal Surgery, Harbin Medical University Cancer Hospital, Harbin 150081, China. [6]Department of Gastroenterology, Tianjin Medical University Cancer Institute and Hospital, National Clinical Research Center for Cancer, Key Laboratory of Cancer Prevention and Therapy, Tianjin's Clinical Research Center for Cancer, Tianjin 300060, China. [7]Department of Gastrointestinal Surgery, Harbin Medical University Cancer Hospital, Harbin 150081, China. [8]Shanghai Key Laboratory of Metabolic Remodeling and Health, Institute of Metabolism and Integrative Biology, Institutes of Biomedical Sciences, Fudan University, Shanghai 200032, China. [9]Shanghai Qi Zhi Institute, Shanghai 200438, China. [10]Department of Oncology Surgery, Harbin Medical University Cancer Hospital, Harbin 150081, China. [11]Key Laboratory of Tumor Immunology in Heilongjiang, Harbin 150081, China. [12]These authors contributed equally: Yangzi Chen, Bohong Wang, Yizi Zhao, Xinxin Shao, Mingshuo Wang, Fuhai Ma. ✉e-mail: tianyantao@cicams.ac.cn; leospiv@hrbmu.edu.cn; zeping_hu@tsinghua.edu.cn

Both genetic and environmental risk factors of GC lead to metabolic changes and further contribute to tumor initiation and progression[10,11]. As a systematic analysis, metabolomics offers a comprehensive profile of metabolic status, reflecting the net result of genetic-environmental interactions. Consequently, it has been widely used to decipher metabolic differences, uncover biomarkers, and identify potential therapeutic targets in various diseases[12–18]. Previous studies have elucidated the application of metabolomics in GC to uncover molecular mechanisms, identify prediction biomarkers for GC occurrence, prognosis, and peritoneal recurrence[19–31]. However, a majority of these studies encountered limitations, including small cohort sizes, the absence of an independent cohort for validation, restricted reproducibility attributed to differences in sample types and detection methods, and low detection sensitivity associated with the analytical techniques employed. Consequently, the development and refinement of metabolic biomarkers suitable for clinical applications remain imperative. Therefore, global metabolomic profiling of multiple large, well-characterized cohorts is crucial for identifying and validating biomarkers with translational potential.

Although metabolomics enables the measurement of hundreds of metabolites presenting in clinical samples, sophisticated data processing and interpretation remain a challenge. Machine learning, a widely used artificial intelligence (AI) approach, automatically analyzes complex data in many fields of biomedical science[32,33]. It presents unique advantages, especially in interpreting -omics data, developing prediction models, identifying biomarkers, and stratifying patients for precision medicine[34–39]. However, the utilization of machine learning in analyzing GC metabolomics data and developing potential biomarkers is underexplored, highlighting substantial potential for further research.

Here, liquid chromatography-mass spectrometry (LC–MS)-based targeted metabolomics was employed to analyze plasma samples from multi-center GC patients and non-GC controls (NGC), totaling 702 participants. A diagnostic model was developed by machine learning using metabolomics data and was further validated in both test set 1 and test set 2. Notably, this diagnostic approach outperformed the conventional method that utilized cancer protein markers including Carbohydrate Antigen 19-9 (CA19-9), Cancer Antigen 72-4 (CA72-4), and Carcinoembryonic Antigen (CEA) in identifying patients at stage IA and other stages. In addition, the prognostic biomarker panel showed a remarkably higher concordance index (C-index) than the traditional method that employed clinical indications, suggesting better performance in predicting clinical outcomes. Moreover, the model-based risk stratification of patients could inform clinical decision making. Collectively, our study presented empirical findings demonstrating the advantages of applying machine learning in analyzing metabolomics data for enabling early detection and precision medicine in GC.

## Results

### Patients, data collection, and study design

The overall workflow of this study and the detailed participant recruitment information for each analysis were illustrated in (Fig. 1). Specifically, plasma samples were obtained from a total of 702 individuals consisting of 389 GC patients and 313 NGC. The clinical characteristics of those participants were summarized in Supplementary Fig. 1a–d. Next, the metabolomics profile of plasma samples was obtained using a targeted liquid metabolomics approach based on LC–MS[15]. In total, 147 metabolites including amino acids, organic acids, nucleotides, nucleosides, vitamins, acylcarnitines, amines, and carbohydrates were detected (Supplementary Fig. 1e). Then the metabolic landscape of GC and NGC in Cohort 1 were compared and the association between the metabolic signatures and clinical phenotypes was investigated using machine learning algorithms. We developed a GC diagnostic model named the 10-DM model and evaluated the model performance in distinguishing GC patients from NGC. In addition, an

external test set 2 (Cohort 2) was applied to validate the model's robustness. Apart from the diagnostic model, we further constructed a prognostic model (28-PM model) using machine learning analysis of metabolomics data from 181 GC patients (Cohort 3). We also benchmarked the model performance against traditional methods that leverage clinical indications and assessed the risk-stratification ability of the model.

### Reprogrammed plasma metabolic landscape in GC patients

To characterize the plasma metabolic reprogramming of GC, metabolomic analysis was performed in GC patients versus NGC. Specifically, a principal component analysis (PCA) distinguished GC from NGC samples, indicating that GC metabolome undergoes remodeling (Fig. 2a). In total, 45 metabolites were statistically different in GC compared against NGC (Wilcoxon rank-sum test, false discovery rate (FDR) < 0.05 and fold change > 1.25 or < 0.8) (Fig. 2b and Supplementary Fig. 2a–b). Interestingly, these dysregulated metabolites showed 3 remarkably distinct trends (Cluster 1–3) along with the disease progression (Fig. 2c and Supplementary Fig. 2c–e). Particularly, the metabolites in Cluster 1 (e.g., neopterin and N(7)-methylguanosine) exhibited a sustainable increasing pattern while those metabolites in Cluster 2 (e.g., glutathione disulfide (GSSG), uridine, and lactate) showed a continuously decreasing trend along with cancer initiation and progression (Fig. 2c and Supplementary Fig. 2c, d).

Furthermore, KEGG pathway enrichment analysis of these differential metabolites revealed a range of disturbed metabolic pathways (Fig. 2d). Glutathione metabolism, which has been well characterized previously in several cancers with functions in the cellular antioxidant system, reactive oxygen species management and the potential in anticancer therapeutics[40,41], was the most significantly disturbed pathway in GC. Two key metabolites in the glutathione metabolism, GSH, and GSSG, were significantly decreased in the GC plasma (Supplementary Fig. 2a, b). However, the GSH/GSSG ratio, which has been identified as an indicator of disturbed oxidative stress[42–44], was significantly upregulated in GC patients and increased along with disease progression (Supplementary Fig. 2a). Taken together, the data showed that oxidative stress was greatly dysregulated in GC patients.

Additionally, cysteine and methionine metabolism was also vigorously perturbed metabolic pathway in GC patients, which was reported to influence oxidative stress, mediate cellular signaling, and facilitate epigenetic regulation in the tumorigenesis process[45–48]. Moreover, the down-regulation of S-Adenosyl-L-homocysteine (SAH), up-regulation of S-Adenosyl methionine (SAM), and an increasing trajectory of SAM/SAH ratio along with disease progression in GC patients in comparison with NGC controls were observed (Supplementary Fig. 2a, b). As a universal methyl donor, SAM abundance alteration leads to epigenetic changes and regulates gene expression, supporting cell proliferation and growth[49–52]. Therefore, the dysregulation of the SAM/SAH ratio may reflect the perturbation of the methyl pool in GC patients.

Together, our findings depicted the metabolic vulnerabilities and underlay potential applications of plasma metabolites in the detection and prediction of GC.

### Biomarker panel derived from machine learning enables GC patient diagnosis at early stages

We next leveraged the reprogrammed metabolic profiles we acquired to develop innovative cancer diagnostic approaches. Machine learning was used to develop a model for predicting the clinical status in this study. Using the Least Absolute Shrinkage and Selection Operator (LASSO) regression algorithm, we selected 10 essential metabolites for the discrimination of GC and NGC (Fig. 3a), including succinate, uridine, lactate, SAM, pyroglutamate, 2-aminooctanoate, neopterin, N-Acetyl-D-glucosamine 6-phosphate (GlcNAc6p), serotonin, and nicotinamide mononucleotide (NMN). Next, we trained a random

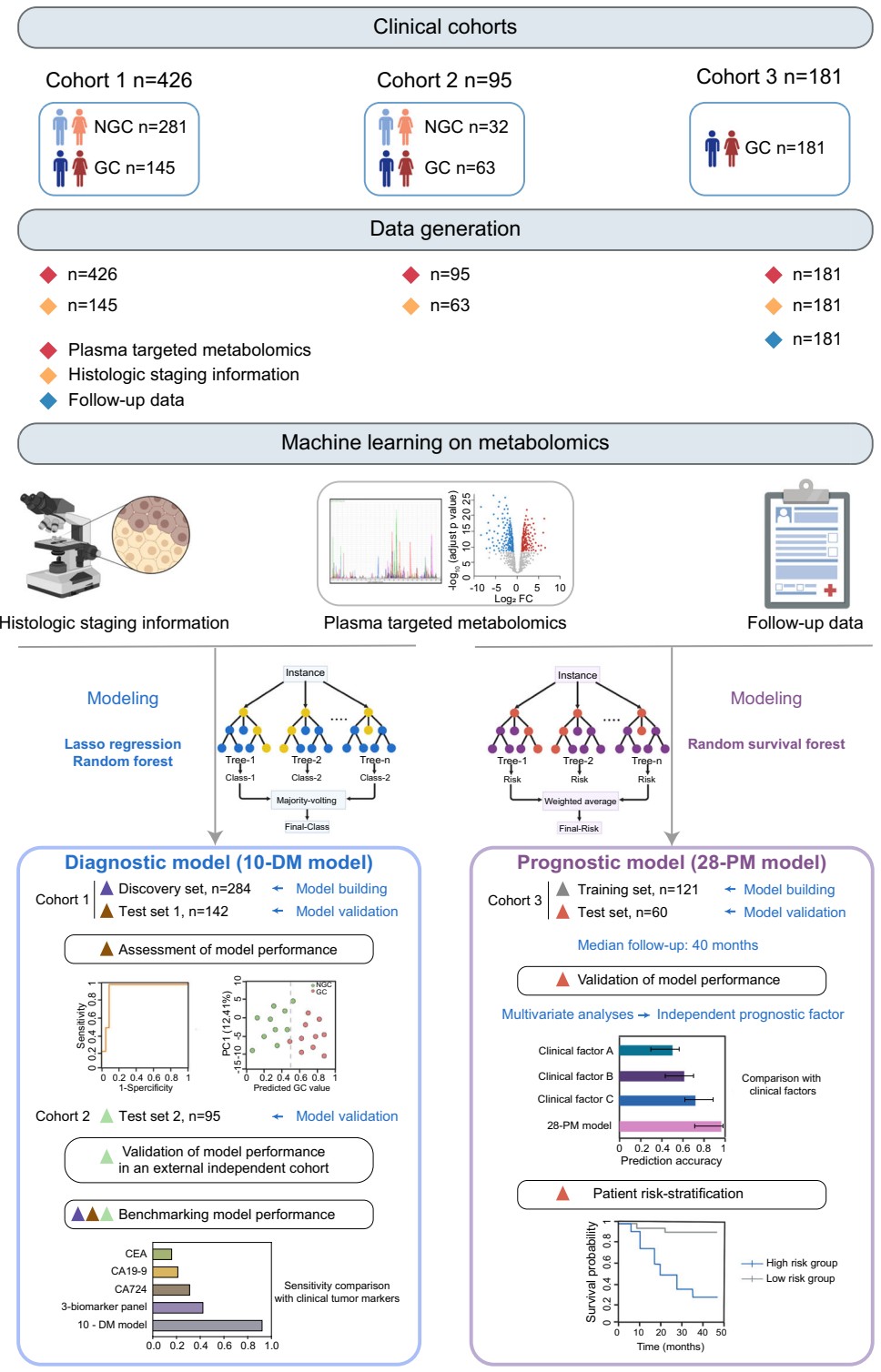

**Fig. 1 | Schematic overview of the study.** Overview of the study design. The illustration was created with a full license on BioRender.com. A total of 702 individuals were included in the study, and their plasma samples underwent targeted metabolomics analysis. The metabolic profiles of gastric cancer (GC) patients and non-GC controls (NGC) in Cohort 1 ($n$ = 426) were compared to depict the metabolic reprogramming in GC. Using the metabolomics data from Cohort 1 and machine learning techniques, a diagnostic model for GC (10-DM model) was created and validated. This model was further verified in the test set 2 (Cohort 2, $n$ = 95). Metabolomics data from Cohort 3 ($n$ = 181) patients and their clinical features were analyzed using a machine learning algorithm to develop a prognostic model (28-PM model). The performance of these two models was benchmarked against clinically used biomarkers/clinical features. Different colored triangles in the figure represent various participant groups used for model construction, validation, and comparison processes. Source data are provided as a Source Data file.

forest model with the 10 essential features, and then validated the model in the test set 1, yielding an area under the receiver operating characteristic (AUROC) of 0.967 (95% confidence interval (CI): 0.944-0.987, sensitivity: 0.854, specificity: 0.926) (Fig. 3b). Moreover,

each metabolite contributed relatively evenly to this 10-metabolite diagnostic model (10-DM model), with succinate, uridine, and lactate being the three most significant contributing metabolites (Fig. 3c). Previous studies on gastrointestinal tumors have consistently

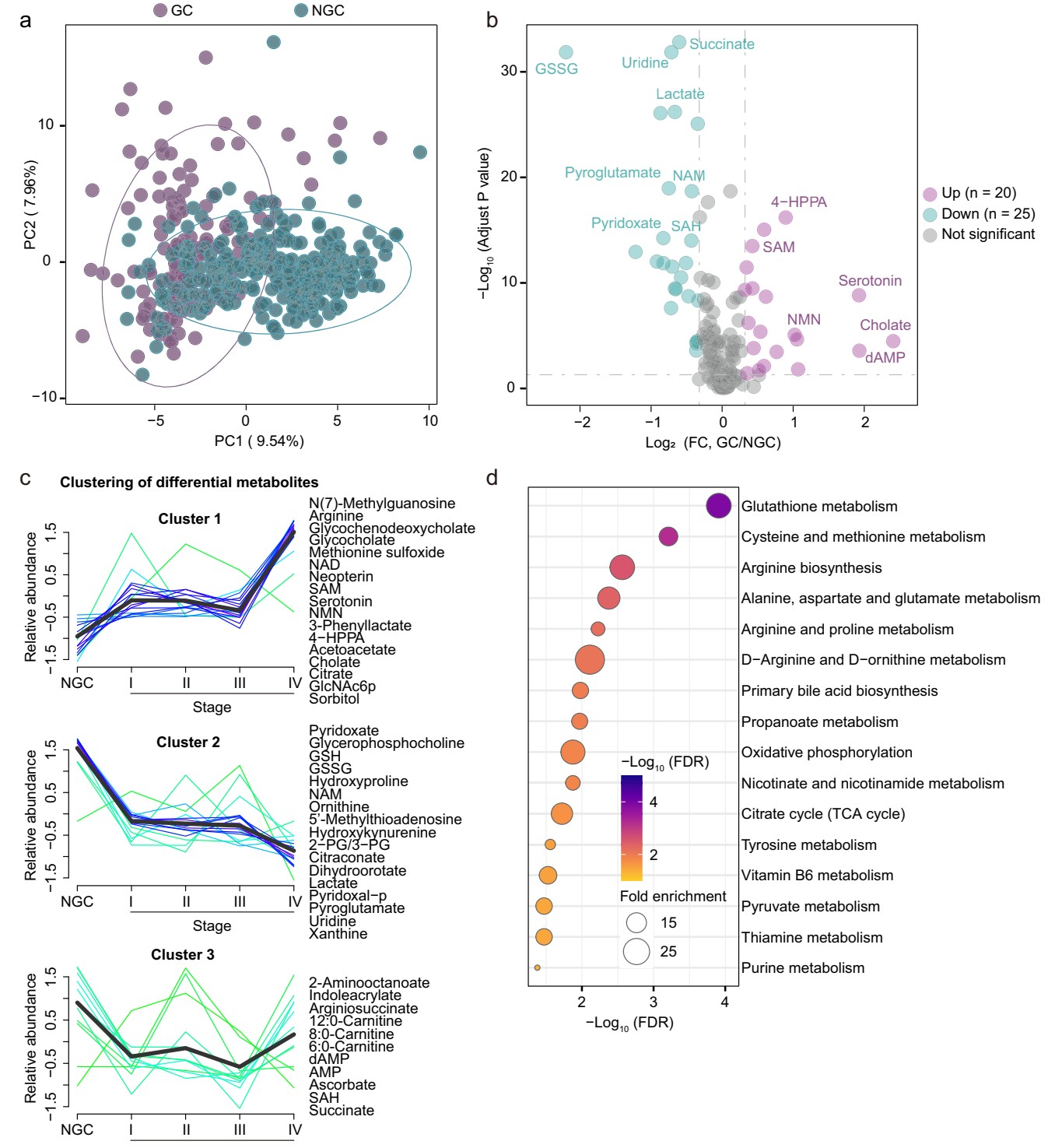

**Fig. 2 | Reprogrammed plasma metabolic landscape of GC patients compared with non-GC controls. a** Principal Component Analysis (PCA) of the Cohort 1 (*n* = 426) plasma-targeted metabolomics data comparing GC patients (colored in purple) and NGC controls (colored in green). **b** Volcano plot of the detected metabolites in Cohort 1 plasma metabolomics (GC patients versus NGC controls). Significantly differential metabolites are colored in purple (upregulated) and green (downregulated); the others are colored in gray. Two-sided Wilcoxon rank-sum test followed by Benjamini–Hochberg (BH) multiple comparison test with false discovery rate (FDR) < 0.05 and fold change (FC) > 1.25 or < 0.8. **c** Mfuzz clustering of metabolic trajectories during GC progression using the differential metabolites according to the metabolic changes' similarity. Representative metabolites of each cluster are presented on the side. **d** Kyoto Encyclopedia of Genes and Genomes (KEGG) metabolic pathways enriched by significantly differential metabolites between GC patients and NGC controls. One-sided Fisher's exact test followed by BH multiple comparison tests was used and only pathways with FDR < 0.05 were presented. Source data are provided as a Source Data file.

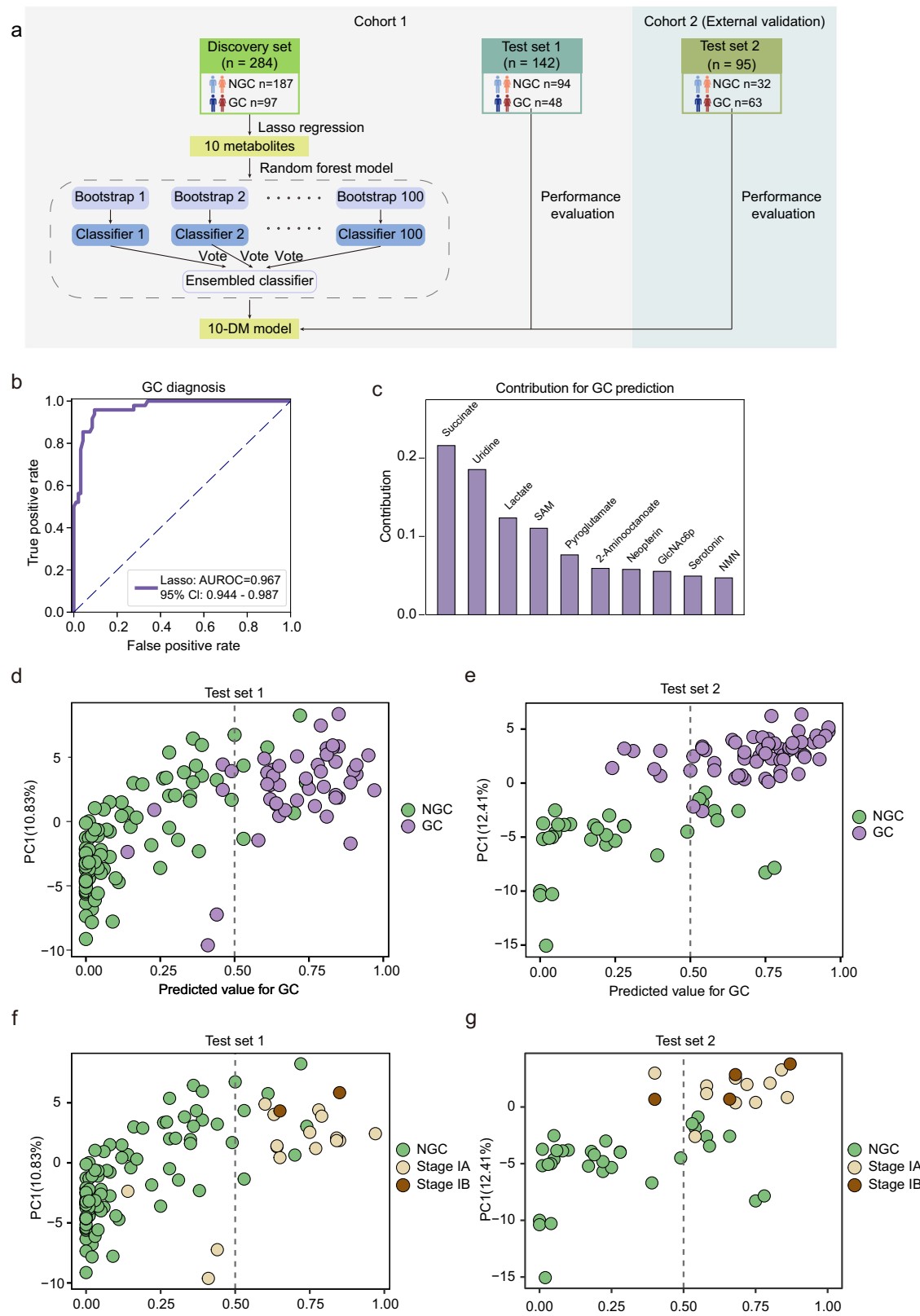

identified differential metabolites, including succinate[53,54], uridine[55], and lactate[24]. Succinate and lactate have been continuously upregulated in the epithelium, serrated lesions, and tumor tissues of GC patients, implying their involvement in tumor initiation and progression[56]. Significant alterations in uridine levels have been detected in GC tumor tissues[55]. Likewise, the relative abundance plots

across the tumor initiation and progression indicated that all of these ten metabolites were significantly different between GC and NGC, with five of them (SAM, neopterin, GlcNAc6p, serotonin, and NMN) being significantly upregulated in GC and the other five (succinate, uridine, lactate, pyroglutamate, and 2-aminooctanoate) significantly downregulated in GC (Supplementary Fig. 3a).

**Fig. 3 | Machine learning-derived prediction model based on plasma metabolome for GC diagnosis. a** Design of the modeling workflow. LASSO regression and random forest algorithm were adopted for feature selection and model training. The 10-DM model was validated in a test set and an external test set. The illustration was created with a full license on BioRender.com. **b** The Receiver operating characteristic (ROC) curve for the diagnosis of GC patients in the test set 1. A 95% confidence interval was calculated based on the mean and covariance of one thousand random sampling tests. **c** Contribution of the ten metabolites to the 10-DM model. **d**–**g**, The prediction performance of the 10-DM model for distinguishing GC (colored in purple) from NGC (colored in green) in the test set 1 (**d**) and the test set 2 (**e**) and for distinguishing stage I GC patients (stage IA colored in yellow and stage IB colored in brown) from NGC in the test set 1 (**f**) and the test set 2 (**g**). The dotted line represented the cutoff value of 0.50 used to separate the predicted NGC (on the left side) from GC (on the right side). Source data are provided as a Source Data file.

To visually demonstrate the model's performance, we generated plots that compare each participant's prediction value with their actual disease status (NGC/GC). Employing a cutoff value of 0.5 for classification, the 10-DM model accurately identified 85.4% of the test set 1 GC patients and 90.3% of the test set 2 GC patients (Fig. 3d, e). In clinical practice, the early detection of GC is crucial for timely clinical intervention and curative resection, which can significantly improve the survival rate of tumor patients[37,57,58]. To further assess the effectiveness of our model in diagnosing early-stage GC, we applied the 10-DM model to distinguish between stage IA/IB GC and NGC in test set 1. The model achieved a prediction accuracy of 90.9% (AUROC: 0.957, 95% CI: 0.917–0.990, sensitivity: 0.813, specificity: 0.926) for stage IA patients and a prediction accuracy of 0.927 (AUROC: 0.984, 95% CI: 0.947–1.000, sensitivity: 1, specificity: 0.926) for stage IB patients, demonstrating its superior discrimination ability in screening early-stage patients (Fig. 3f). Additionally, in the external test set 2 (Cohort 2), the model replicated its performance with an AUROC of 0.920 (sensitivity:0.905, specificity:0.75). Consistent with the previous encouraging results, 83.6% of the early-stage (stage I and stage II) patients in test set 2 were correctly identified by the 10-DM model (sensitivity: 0.931, specificity: 0.75) (Fig. 3g and Supplementary Fig. 3b), and the 10-DM model's detection accuracy for stage IA patients was 79.1% (AUROC: 0.909, 95% CI: 0.838–0.975, sensitivity:0.909, specificity:0.75), indicating its high sensitivity and reliability.

## Comparison of the diagnostic performance of the 10-DM model with traditional methods using routine biomarkers and models employing other algorithms

To assess whether the 10-DM model exhibits advance in the diagnosis, we benchmarked the 10-DM model's prediction accuracy against that of the 3 existing clinical tumor biomarkers CA19-9, CA72-4, and CEA (collectively named 3-biomarker panel). The discriminative sensitivities of the CA19-9, CA72-4, and CEA were 0.217, 0.317, and 0.165 respectively, compared to 0.925 of the 10-DM model (Supplementary Fig. 4a, b). Considering that these three biomarkers are frequently combined in clinical practice to enhance specificity, we hypothesized that sensitivity could be improved if we classified an individual as a GC patient if any single metabolite of the 3-biomarker panel falls outside the normal range (i.e., CEA: 0–5 µg/L, CA19-9: 0–27 U/mL, CA72-4: 0–6.9 U/mL). Strikingly, our 10-DM model showed superior performance even over the 3-biomarker panel (sensitivity 0.925 versus 0.428) (Supplementary Fig. 4b). It should be noted that the better performance of the 10-DM model was not an artifact from high false positive rate (Fig. 3b, d, e). The integration of the three biomarkers improves the sensitivity of the 10-DM model (from 0.925 to 0.957) (Supplementary Fig. 4b), suggesting the potential to enhance the applicability of the 10-DM model in current clinical practices.

Moreover, we also benchmarked the performance of the 10-DM model with different machine learning algorithms in Metaboanalyst including Support Vector Machine (SVM), Logistic Regression (LR), Random Forest (RF), and PLS-DA. The 10-DM model consistently demonstrated the best model performance (Supplementary Fig. 4c).

Together, our data demonstrated that the 10-DM model provided significantly higher accuracy than the conventional 3-biomarker panel routinely used in clinical practice and other algorithms in Metaboanalyst for the detection of GC patients.

## The metabolic prognostic model accurately predicts GC patient outcomes

As precise prognosis could enable precision intervention and benefit the treatment outcome of the patients clinically, we also attempted to develop a machine learning-derived prognostic model. To this end, we collected the metabolomics profiles in plasma from 181 GC patients (Cohort 3) and gathered their clinical information with a median follow-up period of 40 months. Then we established a 28-metabolite prognostic model (28-PM model) by using the random survival forest method. Specifically, the training set patients were involved in the model construction using 147 metabolites initially. Then, to avoid model overfitting, 28 metabolites were selected as key features for re-training an optimal model (28-PM model) with a concordance index (c-index) of 0.90 (Fig. 4a and Supplementary Fig. 5a). Afterwards, the 28-PM model was evaluated on the test set, showing effective predictive power, achieved an AUROC of 0.832 (95% CI: 0.697–0.951, sensitivity: 0.900, specificity: 0.700) and a c-index of 0.83 (Fig. 4b). Interestingly, We observed that only 11 of the 28 metabolites' relative abundance could significantly distinguish the overall survival of test set patients, including symmetric dimethylarginine/asymmetric dimethylarginine (SDMA/ADMA), neopterin, thymine, glucuronate, hydroxyproline, 14:0 Carnitine, indoleacrylate, 8:0 Carnitine, acetylalanine, 2-aminoadipate, and GlcNAc6p (Supplementary Fig. 5b). ADMA promotes the migration and invasion of gastric cancer cells through enhancing epithelial-mesenchymal transition (EMT) and regulating β-catenin expression in GC[59]. Elevated levels of 14:0 carnitine and 8:0 carnitine were associated with a worse outcome. Previous studies on GC have identified increased expression of CPT1, the rate-limiting enzyme regulating long-chain fatty acid oxidation, accelerating GC progression. The expression levels of CPT1C could also affect the outcome of GC patients. Moreover, the role of CPT1 in other cancers has also been reported, suggesting that fatty acid metabolism might play a vital role in cancer metabolic adaptation[60–63]. In addition, elevated levels of neopterin were indicative of a poor prognosis. Neopterin is produced by macrophages or DC cells stimulated by IFNγ, commonly regarded as one of the biomarkers for immune activation[64]. In a single-cell transcriptomic study of GC, it was found that macrophages in tumor microenvironment play multiple roles in modulating tumor immunity[65]. Furthermore, neopterin has been demonstrated in various studies to possess the potential capability for prognosis monitoring including endometrial cancer, prostate cancer, colorectal cancer, and gastric cancer[66–69], which might explain the elevated plasma levels of neopterin. Together, our machine learning-derived prognostic model showed good performance in predicting the clinical prognosis of GC patients.

## The addition of clinical parameters barely strengthened the prognostic capability of the 28-PM model

To assess the predictive prowess of our model in comparison to clinical factors employed by clinicians for empirical prognostic assessment, we initially conducted a screening of clinical variables associated with prognosis using univariate Cox regression analysis. We identified

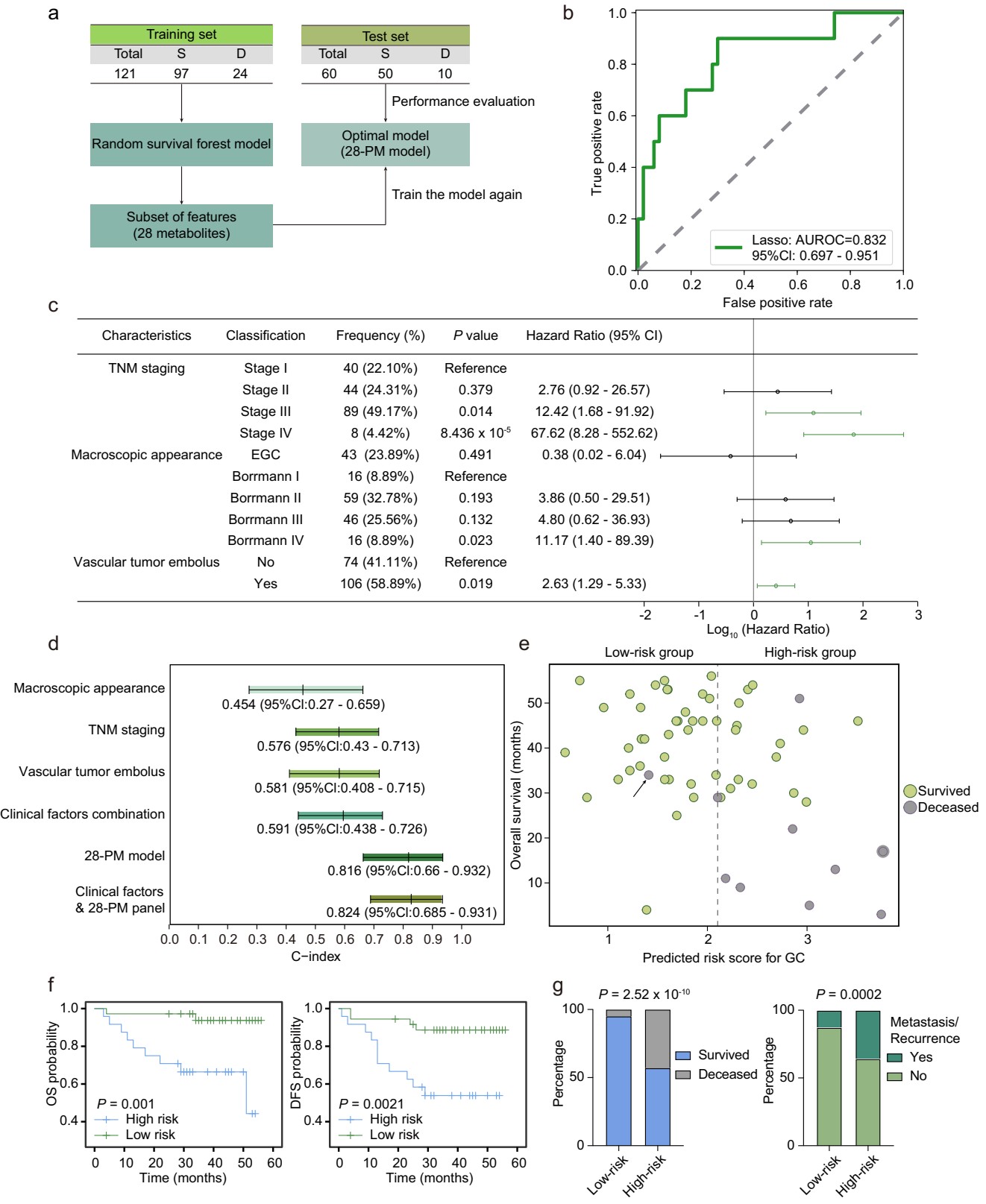

TNM staging, macroscopic appearance, and vascular tumor embolus as three clinically relevant factors significantly correlated with prognosis ($P < 0.05$) (Fig. 4c and Supplementary Table 1). Subsequently, through a comparative analysis utilizing C-index values as indicators of model performance, we determined that the predictive efficacy of each of these three clinical factors, whether considered individually or in combination, was inferior to that exhibited by the 28-PM model. This

observation underscores the superior predictive capability of our model relative to traditional clinical factors. Considering the influence of clinical indicators on prognostic prediction, we further attempted to incorporate a combination of clinical characteristics into the 28-PM model to assess whether this would enhance the predictive capabilities of the 28-PM model. As illustrated in Fig. 4d and Supplementary Fig. 5c, the metabolic model 28-PM exhibits greater robustness in predicting

**Fig. 4 | The prognostic model outperformed clinical parameters in predicting outcomes of GC patients. a** Schematic outline of the prognostic model design. S survived, D deceased. **b** ROC curve analysis of the test set. 95% CI was calculated based on the mean and covariance of one thousand random sampling tests. **c** Forest plot of clinical parameters with significant prognostic relevance identified by univariate Cox regression analysis. Parameters with a $P < 0.05$ were considered statistically significant and represented by green lines. The center dots and lines represent HR and 95% Cl scaled by log 10. EGC, early gastric cancer. *P*-values of TNM staging, macroscopic appearance, and vascular tumor embolus were calculated based on data from $n = 181, 180$, and 180 independent samples respectively. **d** C-index values comparison of the macroscopic appearance, TNM staging,

vascular tumor embolus, and the 28-PM model in the test set ($n = 60$). C-index and the 95% Cl were presented under the relative colored bars. **e** Prognostic prediction of the test set patients ($n = 60$) using the 28-PM model. The dotted line drawn at the cutoff value of 2.1 divided the patients into high- and low-risk groups. Green circles and gray circles represent survived and deceased in the test set. The arrow pointed out the deceased patient dying of a heart attack. **f** Kaplan–Meier curves showing the overall survival (OS) and disease-free survival (DFS) of test set GC patients ($n = 60$) stratified by prognostic risk scores (cutoff = 2.1). *P*-values were calculated with a two-sided log-rank test. **g** The high-risk group presented a higher proportion of deceased and relapse/metastasis. A two-sided Fisher's exact test was used to calculate the *P*-value. Source data are provided as a Source Data file.

GC patients' prognosis among different stages. The metabolic model that integrates clinical features achieves a higher prognostic prediction accuracy for early-stage patients compared to late-stage patients (C-index value 0.868 vs. 0.778). In summary, the incorporation of clinical characteristics into the metabolic model does not yield a substantial improvement in model performance (resulting in only a 1% benefit compared to the 28-PM model).

Afterward, we evaluated the prediction performance of the 28-PM model for each patient in the test set. According to the algorithm-determined cutoff value (see "Methods" section), we stratified the GC patients into a high-risk group and a low-risk group and noted that almost all the deceased belonged to the high-risk group except one patient (Fig. 4e, an arrow pointed) who died of a heart attack, underlying the prognostic capability of the 28-PM model. With the observation that the high-risk patients showed poorer disease-free survival (DFS) and overall survival (OS) compared with the low-risk individuals (Fig. 4f), we further characterized the two groups with the distribution of living status and the recurrence/metastasis circumstances. As expected, the high-risk group exhibited a higher proportion of deceased individuals and the non-metastasis/non-recurrence patients were more prominent in the low-risk group (Fig. 4g), indicating that the 28-PM model successfully identified the patients who need refined therapy regimen. A multivariate Cox regression was performed to demonstrate that the 28-PM model is an independent prognostic factor (Table 1). This outcome signifies our success in developing an accurate method for independently predicting patient prognosis.

**Table 1 | Multivariate Cox regression of GC patients' prognosis in Cohort 3**

| Characteristics | Classification | P-value | Hazard ratio | 95% Cl |
|---|---|---|---|---|
| TNM staging | I | 0.010 | Reference | |
| | II | | 1.19 | 0.07–20.88 |
| | III | | 4.02 | 0.24–66.59 |
| | IV | | 12.86 | 0.70–237.08 |
| Macroscopic appearance | Borrmann I | 0.789 | Reference | |
| | Borrmann II | | 3.69 | 0.47–28.78 |
| | Borrmann III | | 3.35 | 0.43–25.97 |
| | Borrmann IV | | 3.45 | 0.41–29.25 |
| | EGC | | 1.96 | 0.06–60.26 |
| Vascular tumor embolus | No | 0.463 | Reference | |
| | Yes | | 1.38 | 0.59–3.24 |
| Metabolic risk factor* | Low | $5.276 \times 10^{-5}$ | Reference | |
| | High | | 7.53 | 2.83–20.04 |

Multivariate Cox regression was applied to the 28-PM model and clinical parameters to identify independent prognostic factors. Parameters with $P < 0.05$ are recognized as statistically significant, signifying their role as independent prognostic factors for GC. *P*-values were calculated based on data from $n = 179$ independent patient samples and Wald test.
*CI* confidence interval, *EGC* early gastric cancer.
*The classifier was derived from the 28-PM model cutoff value, represents as a high-risk group and a low-risk group for the GC patients.

Together, our study provided a more accurate model-driven approach for prognostic prediction and clinical decision making which could be easily implemented in routine patient care.

## Discussion

In this study, we employed multi-center clinical cohorts to investigate the metabolic alterations in plasma between GC patients and NGC controls and to identify circulating metabolites with potential diagnostic and prognostic value. Specifically, machine learning algorithms were utilized to analyze the metabolomics data, further developing two biomarker panels termed the 10-DM model and 28-PM model with superior accuracy in comparison with existing clinical methods for GC detection and prognostic prediction, respectively. Collectively, our study demonstrated the unique advantages of applying machine learning-based metabolomics in facilitating the early detection and precision medicine of GC, thereby providing the clinical translation potential in the future.

GC is one of the most common lethal tumors in the world since patients are frequently detected at an advanced stage[1,4]. Therefore, more efforts in developing non-invasive early screening methods and refining precision medicine guidance to prolong patients' survival are urgently needed. To date, several omics studies exploring GC characteristics have been reported[19,70–73], the majority of which, however, have focused on the potential of DNA, RNA, and protein as GC biomarkers[70,74–76]. Considering the tight association between GC and metabolism, our work emphasized the predictive value of metabolites in GC. Metabolomic profiling enables the measurement of hundreds of metabolites, thus could uncover as many potential biomarkers as possible[11–13]. Moreover, we leveraged LC–MS-based metabolomics, which has achieved competitive analytical reproducibility and reliability, to depict the global metabolic remodeling in GC and produced two predictive models (10-DM model and 28-PM model). In the future, optimized targeted metabolomic assays could be established to measure the specific subsets of metabolites in the two models to increase efficiency and reduce costs. In addition, commercialized kits, easy-access devices, and simplified instruments could be developed based on the two prediction models to assist GC early detection and to inform clinical decision making based on the risk stratification of patients and consequently, facilitating precision medicine and clinical translation.

Although metabolomics presents unique advantages in determining GC global metabolic signatures and can identify promising biomarkers for GC diagnosis and prognosis, the interpretation of the sophisticated -omics data has always been a challenge. In the past few years, machine learning algorithms have been employed in discovering the underlying association between the -omics data and disease status and creating predicting models[6,32,33]. Accordingly, we leveraged machine learning algorithms in our study and proved its capability from three aspects. First, for purposes of avoiding overfitting, the LASSO regression algorithm was implemented in the selection of essential features among the total 147 metabolites before modeling the 10-DM model. Second, compared with previous biomarker-discovery studies which employed logistic regression algorithms to

train the model, we chose the random forest and the random survival forest algorithm to obtain higher robustness and less overfitting to train the diagnostic model and the prognostic model respectively[32,33]. Third, the algorithms successfully revealed the prediction potential of some metabolites which tend to be ignored or missed by the traditional analytical ways. For instance, the metabolites of the 10-DM model exhibited moderate alteration between GC and NGC. A possible explanation was that machine learning algorithms unveiled the latent rules in the complex correlations among metabolites and picked the most representative metabolites in distinguishing GC and NGC. Another example demonstrating machine learning's advantage in uncovering potential biomarkers was the 28-PM model. As mentioned before, only 11 metabolites in the 28-PM model were independently relative to the patient's outcomes (Supplementary Fig. 5b). However, these 11 metabolites failed to train a prognostic model, indicating the prognostic prediction ability of the other 17 metabolites and emphasizing the necessity of hiring machine learning algorithms to uncover the comprehensive association between metabolism and prognostic status.

Both the diagnostic and prognostic models we created employing machine learning algorithms were validated of generalization and present superior performance than traditional methods leveraging clinical existing factors. Specifically, our data demonstrated that the diagnostic model (10-DM model) was of particular importance as it accurately identified GC patients even at stage IA, dramatically outperforming the existing clinical markers. Notably, the accuracy and reproducibility of the 10-DM model were confirmed in multi-center cohorts covering 521 individuals, indicating the high robustness and clinical application potential of the model. Apart from the diagnostic model, we further defined and validated the prognostic panel termed the 28-PM model. This model outperformed the clinical parameter combination, demonstrated by a much higher C-index value. Ultimately, high-risk patients distinguished by the 28-PM model showed poorer outcomes compared with low-risk patients, suggesting the forceful prognostic capability of the model. After the model-guided patients' stratification, precise clinical decisions could be made for the individuals. Patients who are stratified to the high-risk group are more likely to benefit from intensive monitoring, prompt intervention, and trials of therapeutic agents. In the future, the enlarged participants' scale and updated machine learning algorithms could be leveraged to verify and optimize the two models further for clinical translational research.

In summary, the strengths of our study include a large-scale, multi-center clinical cohorts-based highly sensitive targeted metabolomics analysis of GC patients and NGC controls, which depicts the metabolic reprogramming landscape of GC patients and provides a valuable resource that expands our knowledge of the GC. Moreover, the application of machine learning and metabolomics presenting remarkable advantages is complementary to a range of studies surrounding GC characterization and precision medicine. In addition, the two models determined in our study were constructed based on a simple set of metabolites, facilitating replication, optimization, and clinical application. In the future, the two models, by assessing of relative range of metabolites, could be used in various situations without being constrained by tools and detection techniques.

The limitations of our study should also be noted. The Cohort 3 patients' median follow-up time of 40 months was insufficient (ideally, it should have been more than 5 years), which could have resulted in inadequate observation. A longer follow-up duration may aid in a more thorough evaluation and optimization of the 28-PM model. At the conclusion of the follow-up period, a significant number of patients had right-censored data. Despite our use of Random Survival Forest to address this limitation, a larger patient cohort would offer the opportunity to collect more complete data (where the survival time of each patient is unambiguously observed), thereby improving the predictive

model. Moreover, we were not able to provide clinical intervention regimens for the prognosis model. Since there was no statistically significant difference in the proportions of cMET amplification, dMMR, and Laurén types between the two groups (Supplementary Data 6), it is likely that treatments that target these characteristics are not yet appropriate for model-stratified patients. Further characterization of the high- and low-risk patients categorized by the model is needed in the future to inform treatment regimens. It will also be fascinating to investigate whether the responses to chemotherapy and immunotherapy in high-risk and low-risk patients are different, which could help guide appropriate therapeutic strategies.

It's essential to understand that current models are not yet appropriate for direct application in clinical settings due to the following limitation. Given that the model is constructed based on relatively quantitative metabolomics data, understanding the GC prognosis or risk regarding to new patients using these machine learning (ML) models necessitates the concurrent presence of quality control (QC) samples employed during the ML model construction process. However, our study identified crucial metabolites capable of distinguishing between GC and NGC, representing a significant step towards constructing a model with potential clinical applications. In the future, to further advance the translation of our research into clinical practice, we intend to conduct absolute quantitative metabolomics with large-scale multi-center patient samples using isotopic internal standards of these key metabolites. This will help to elucidate the normal range and problematic range of the important metabolites, thereby determining detection thresholds for GC. Additionally, we will explore alternative detection methods for differential metabolites, including simplified mass spectrometry detection methods and novel detection strategies such as assay kits, aiming to streamline detection time and costs and facilitate clinical applications.

Collectively, our discoveries delineated metabolic reprogramming in GC and incorporated machine learning algorithms to construct two models that identify GC patients and predict their prognosis, respectively. Our work enhanced the understanding of GC pathology, facilitated the development of GC early detection, and shed light on the precision treatment of GC. More generally, the framework highlights the unique advantages of machine learning-based -omics data interpretation for tumor detection and decision guidance and could be generalized to explore other diseases.

## Methods
### Patient characteristics
A total of 389 patients with pathologically confirmed GC and 313 non-GC controls were recruited from September 2017 to December 2022.

Cohort 1 was obtained from the Harbin Medical University Cancer Hospital and the National Cancer Center/National Clinical Research Center for Cancer/Cancer Hospital, Chinese Academy of Medical Sciences, and Peking Union Medical College from June 2022 to October 2022. Characteristics of Cohort 1 participants ($\bar{i} = 426$): 145 GC patients (median age 60 years, 71% male, Stage I/II/III/IV, 52/30/53/10); 281 NGC (median age 50 years, 52% male).

Cohort 2 was collected from Harbin Medical University Cancer Hospital (October 2022 to December 2022). Characteristics of Cohort 2 participants ($n = 95$): 63 GC patients (median age 64 years, 76% male, Stage I/II/III/IV, 16/14/24/9); 32 NGC (median age 51 years, 88% male).

Cohort 3 was recruited from the National Cancer Center/National Clinical Research Center for Cancer/Cancer Hospital, Chinese Academy of Medical Sciences, and Peking Union Medical College from September 2017 to October 2022. Characteristics of Cohort 3 participants ($n = 181$): median age 60 years, 63% male, Stage I/II/III/IV, 40/44/89/8.

Patients receiving anti-cancer treatments before sampling were excluded. The study complies with all relevant ethical regulations and was approved by the Research Ethics Committee of the National

Cancer Center/National Clinical Research Center for Cancer/Cancer Hospital, Chinese Academy of Medical Sciences and Peking Union Medical College (Institutional Review Board: 20/086-2282) and Harbin Medical University Cancer Hospital (Institutional Review Board: KY2018-03, 2019-164-R). Written informed consent was obtained from each individual.

Clinical information including sex, age, BMI, tumor pathological variables, and prognostic status was collected and presented in Supplementary Fig. 1.

## Plasma sample collection
Peripheral venous blood-derived plasma samples were collected from preoperative GC patients and non-GC controls in Cohort 1 and Cohort 2. For Cohort 3 patients, preoperative superior vena cava blood-derived plasma samples were gathered. After an overnight fast, blood was drawn using BD Vacutainer EDTA tubes and processed for plasma isolation following the procedures as described below: The blood was centrifuged at $1000 \times g$ for 10 min at 4 °C. The supernatant was collected and centrifuged at $2000 \times g$ for 5 min at 4 °C. Plasma was frozen at −80 °C until metabolite extraction.

## Metabolite extraction
For metabolite extraction of plasma samples, 40 µL plasma of each person was mixed with 160 µL ice-cold methanol. The mixture was then vortexed for 1 min and centrifuged at 4 °C for 15 min with a speed of 15000 rpm. The supernatant was collected and divided into 2 replicates for evaporating in a speed vacuum concentrator. The dried metabolomic samples were kept at −80 °C until LC–MS analysis. For quality control (QC) samples, 10 µL plasma of each person was mixed and then processed the same as that of the study plasma samples.

## Targeted metabolomics analysis
The targeted metabolomics analysis was performed similarly to our previous work. In brief, the dried metabolites were reconstituted in 50 µL water with 0.03% formic acid. After being vortex-mixed vigorously, the samples were centrifuged to remove debris at 4 °C for 15 min with a speed of 15,000 rpm. Before LC–MS/MS, samples were randomized and blinded to avoid the impact of instrument fluctuations on the results. Chromatographic separation was performed on an RP-UPLC column (HSS T3, 2.1 mm × 150 mm, 1.8 µm, Waters) with the following gradient: 0–3 min 99% A; 3–15 min 99–1% A; 15–17 min 1% A; 17–17.1 min 1–99% A; 17.1–20 min 99% A. Mobile phase A was 0.03% formic acid in water. Mobile phase B was 0.03% formic acid in acetonitrile. The flow rate was set as 0.25 mL/min, and the injection volume was 10 µL. The column temperature and autosampler temperature were set to 35 °C and 4 °C, respectively. Mass data acquisition was performed using an AB QTRAP 6500+ triple quadrupole mass spectrometer (SCIEX, Framingham, MA) in multiple reaction monitoring (MRM) mode to monitor 258 unique endogenous water-soluble metabolites[15]. Chromatogram review and peak area integration were performed using MultiQuant 3.0.2 (SCIEX, Framingham, MA). The processed data were exported for further analysis. The missing value was removed according to the 80% rule[77], wherein, a metabolite was considered as detectable when it was detected across at least 4/5 samples in one group. Following this rule, 147 metabolites were robustly detected across all the 702 samples with a small proportion of undetected metabolites (<1/5 samples) were filled with a detection baseline value, 1000, to allow the following statistical analysis.

## Data analysis and preprocessing
QC samples were inserted in an interval of ten test samples to monitor the stability of the instrument and normalize the variations during the run. Therefore, it can serve as an additional QC measure of analytical performance and a reference for normalizing raw metabolomics data across samples.

The detailed process was performed following the procedures as described below[15]. Briefly, the mean peak area of each metabolite from all QC samples in all given batches (QCall), as well as the mean peak area of each metabolite from the QC samples that were the most adjacent to a given group of test samples (QCadj) were first calculated. The ratio between these two mean peak areas for each metabolite was computed by dividing the same QCall by each QCadj and used as the normalization factor for each given group of test samples. The peak area of each metabolite from each test sample was normalized by multiplying their corresponding normalization ratio to obtain the normalized peak areas to remove potential batch variations. In addition, to effectively correct the sample-to-sample variation in biomass that may contribute to systematic differences in metabolites abundance detected by LC–MS, we generated the scaled data by comparing the normalized peak area of each metabolite to the sum of the normalized peak area from all the detected metabolites in that given sample.

## Metabolic differential analysis
All Cohort 1 participants, 426 cases in total, including 145 GC patients and 281 NGC, were used for differential metabolic analysis. Differential metabolites were analyzed by a two-sided Wilcoxon rank-sum test (FDR < 0.05 and fold change > 1.25 or < 0.8). Clustering of differential metabolites was conducted by R package 'Mfuzz' (v2.56.0). KEGG pathway enrichment analysis based on significantly differential metabolites was assessed by the R package 'clusterProfiler' (v3.14.3). KEGG metabolic pathways and related metabolites were downloaded through KEGG API (https://www.kegg.jp/kegg/rest/keggapi.html). Significantly enriched KEGG pathways were determined with Fisher's exact test followed by Benjamini–Hochberg (BH) correction and with FDR < 0.05.

## Diagnostic prediction model
A prediction model for GC diagnosis was built using a random forest algorithm with LASSO feature selection. The participants (Cohort 1, $n = 426$) were randomly stratified sampling into a discovery dataset ($n = 284$) and a test set ($n = 142$). Next, we performed the LASSO regression[78] on the discovery dataset to select a reduced number of features that were able to identify GC patients. We set the coefficient of the L1 constraint as 0.01 and selected ten features with nonzero coefficients based on the misclassification error averaged from 10,000 times of random cross-validation. A random forest model was trained with the ten selected metabolites in the discovery dataset using a bootstrap aggregating approach. The final model included a hundred classifier trees which were built using the split criterion of Gini impurity[79]. For each bootstrap sample, the learning algorithm draws random subsets of features for training the individual decision tree. Decision tree learning employs a divide-and-conquer strategy by conducting a greedy search to identify the optimal split points within a tree. This process of splitting was then repeated in a top-down, recursive manner until all, or the majority of records had been classified under specific class labels. An ensemble method termed Bootstrap Aggregation combined prediction from all individual decision trees to make more accurate predictions than an individual model. Afterward, the diagnostic model was applied to the test set. The predicted value for GC diagnosis was computed as the mean predicted probabilities of the trees in the forest. The class probability of a single tree was the fraction of samples of the same class in a leaf. The final prediction was determined through a voting mechanism, where the model yielded a predicted value (between 0 and 1) for each individual, quantifying the model's uncertainty in prediction. Individuals with predicted values greater than 0.5 would be identified as GC patients by the model, or would be considered as NGC on the contrary.

To compare our model with those readily available and commonly used models, we input the training and testing dataset into the

Metaboanalyst biomarker analysis module to build diagnostic models. In the variable selection step, we selected the top ten metabolites based on Lasso frequency and AUC rank. These metabolites include GSSG, succinate, uridine, 2-PG-3-PG, lactate, uracil, pyroglutamate, SAH, SAM, and R5P. Subsequently, we constructed diagnostic models using various algorithms, including linear SVM, PLS-DA, Random Forests, and Logistic Regression, and compared them with the 10-DM model.

LASSO regression and random forest modeling were performed via the scikit-learn package (v.0.24.1) in Python (v.3.7.4).

## Prognostic model

To establish a GC prognostic model, 181 patients from Cohort 3 with right-censored outcome data (147 participants survived at the end of follow-up and 34 participants died during the follow-up period) were randomly stratified sampling into a training dataset ($n = 121$) and a test dataset ($n = 60$). A random survival forest (RSF)[80] model comprising 1000 trees was trained to select prominent features according to their permutation-based feature importance. The optimal model was established by training the random survival model again with the picked 28 metabolites, showing great predictive power (AUROC = 0.832, 95% CI: 0.697–0.951) for predicting the survival outcomes of GC patients in the test dataset.

In addition, the clinical features that were significantly related to patients' outcomes in the univariate Cox regression analysis including TNM staging, macroscopic appearance, and vascular tumor embolus were utilized separately or as a combination to train RSF models. The integrated RSF model (28 metabolite features along with clinical parameters) was also fitted to predict patients' outcome. Model performance was evaluated on the test dataset by C-index.

For individual outcome prediction, a sample was dropped down each tree in the forest using the split criterion of the log-rank test until it reached a terminal node. Data in each terminal was used to non-parametrically estimate the survival and cumulative hazard function using the Kaplan–Meier and Nelson–Aalen estimators, respectively.

The risk score represented the expected number of events for a particular terminal node, which was estimated by the sum of the estimated ensemble cumulative hazard function. Then, the risk scores were used as evaluation criteria to assess the survival outcomes of GC by ROC curve analysis. The best cutoff was 2.1, determined by the highest true positive rate (0.9) together with the lowest false positive rate (0.3). Patients with a risk score greater than 2.1 would be stratified into a high-risk group, which meant they had a higher risk of poorer survival outcomes. RSF was performed via the scikit-survival package (v.0.17.1) in Python (v.3.7.4).

## Statistical analyses

Statistical analysis methods for metabolomic analysis and modeling evaluation were described in the Results, figure legends, and corresponding "Methods" subsections. Specifically, a two-sided Wilcoxon rank-sum test was used when comparing two groups for unpaired samples. A two-sided Kruskal–Wallis test was used when comparing three or more groups. $P$-values were corrected using the BH method to produce FDR. A $P$-value or DFR of less than 0.05 was considered statistically significant. GraphPad Prism (v.9.0), R (v.3.6.0) software (https://www.r-project.org/), SPSS Statistics 27.0 (IBM), and Python (v.3.7.4) were used to conduct tests.

## Reporting summary

Further information on research design is available in the Nature Portfolio Reporting Summary linked to this article.

## Data availability

Clinical information is provided in Supplementary. Fig 1 and Source data_Supplementary Figs._NCOMMS-23-20271B. The metabolomics data is included in Supplementary Data 1. The statistical analysis of Supplementary Fig.3 is included in Supplementary Data 2. The precision data for relative quantification analysis utilizing the targeted metabolomics analysis method on various sample types (standards, plasma, and 293T) and the assessment data of the targeted metabolomic methodology using isotope-labeled metabolites for methodological evaluation, encompassing matrix effect, recovery rate, and quantitative accuracy, are provided in Supplementary Data 3. The utilization of this method for absolute quantification analysis in a small cohort is presented in Supplementary Data 4. Details on the model's predicted values for all samples is available in Supplementary Data 5. The statistical analysis of the proportions of cMET amplification, dMMR, and Laurén types between the high and low-risk groups is included in Supplementary Data 6. Source data are provided with this paper.

## Code availability

Code in this study is available at https://github.com/Yangzi-Chen2023/GC_NC-Res and https://codeocean.com/capsule/5496369/tree.

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

## Acknowledgements

We thank members of the Hu laboratory for critiquing the manuscript. We thank the Harbin Medical University Cancer Hospital and the National Cancer Center/National Clinical Research Center for Cancer/Cancer Hospital, the Chinese Academy of Medical Sciences, and Peking Union Medical College for providing the biospecimen. Z.H. is supported by grants from the National Natural Science Foundation of China (92057209, 81973355). P.H. is supported by grants from the National Natural Science Foundation of China's General Program (82072640), and Outstanding Youth Project of Heilongjiang Natural Science Foundation (YQ2021H023).

## Author contributions

Y.C. and Z.H. designed the study, and wrote the paper. Y.C. performed data analyses, data interpretation, and metabolite extraction experiments. B.W. performed the metabolomics experiments and data processing. Y.Z. performed the machine learning analysis. M.W. performed metabolite extraction experiments and data analyses. X.S., F.M., L.Y., P.J., H.S., S.L., H.W., P.H., and Y.T. provided the clinical samples and information. M.N. assisted in editing the manuscript. K.Y. assisted in metabolomics experiments. T.Y. assisted in machine learning analysis. Z.H. conceived and supervised the project.

## Competing interests

The authors declare no competing interests.
