## [Peer Review File · Nature Communications]

Metabolomic machine learning predictor for diagnosis and prognosis of gastric cancerReviewers' Comments:

Reviewer #1:

Remarks to the Author:

The authors have performed a targeted metabolomics study on cohorts diagnosed with gastric cancer and created machine learning-based model to enable a plasma-based non-invasive early diagnostic tool. The research question is of interest for precision medicine against gastric cancer and it can help decrease treatment plans/costs. Although, this is very well-structured manuscript with respect to utility and diagnosis capability, there are some shortcomings and improvements that should be considered before accepting the manuscript. Therefore, my feedback is "major revision". Below is detailed feedback:

Page 3, lines 52-54: There are few studies with large cohorts and common metabolites. Although these works, including the current study by the authors, have metabolites that could be triggered by the other diseases/cancer (for instance imbalance level of GSH:GSSH associated with lung cancers, HIV etc and it generally indicates the progression of immune dysfunction or neopterin level in prognostic model which is also seen elevated in diseases such as measles and influenza virus disease). Therefore, it is barely explained biologically how specific are the biomarkers suggested for gastric cancer by author and this claim remains still open and unanswered.

Page 4, lines 68-69: better to rephrase this to: "...validated in two external subsets of test cases test set 1(same cohort) and test set2 (from different cohort)"

Page 5, lines 86-89: it is stated in the page 18, lines 556-558 that 258 unique metabolites were measured while here the authors mentioned that 147 metabolites are measured and used. Which one is the correct number of metabolites that were measured by author's method? Also the list of metabolites, their LOQ and LODs as well as ion transitions for MRM, main adduct ions and retention time should be provided in SIF to verify analytical method validation/limitation.

Page 5, lines 94-96: the prognostic model with 28 variables seem to suffer from few deceased cases (34 deceased against 147 survived) and heavily trained/biased on survival case. If this is the case, any of orthogonal modelling approaches or one-class modelling approach should be adopted. Otherwise, author should provide proof that the prognostic model is unbiased.

Page 5, lines 102-103: the cases are overlapping in the score plot and they are barely distinguished. Please, indicate the variance explained and reinterpret/rephrase the conclusion from PCA analysis.

Pages 5 and 6, lines 101-136: the detected and selected metabolites are required to be discussed in more details from biological perspectives and how specific they are with respect to gastric cancer. It should be stated clearly how common are these metabolites between many other cancers and gastric cancer. For instance, as stated before, GS:GSSH associated is with many other underlying conditions and is not gastric cancer specific. These metabolites could be affected by diet habit, aging or other diseases. What is missing from cohort information is about possible other underlying diseases as well as their nutritional habit. Therefore, the discussion should be reached and states biologically how relevant these metabolites are to the GC and if affected by diet habit or underlying diseases since the ML model will use them to decide about GC positive and false positive discovery rate should be addressed considering these facts.

Pages 6-7, line 138-157: please describe clearly how metabolites such as succinic acid, uridine, lactate etc are varied or affected by GC biologically. How specific these metabolites are in GC positive cases and other diseases and show limit of ML model with respect the elevated level of these metabolites and negative GC. Other confusing case is that looking at Figure 3D, most of these metabolites with exception of Uridine and lactate have wide distribution in their log₂ (relative abundance) and overlaps with non-gastric cancer cases. For instance, considering the stage I, II, III with NGC in SAM metabolite, they have almost similar distribution (percentile 85%). Please, provide p-value and significance test group wise to understand clearly the Figure 3D. Show the level of significance for each pairwise comparison and provide explanations why the overlap between different stages of GC and NGC are closely related. Should another scaling or normalization method to be used?

Page, 7, line 157 (Fig 3d) and page 26, Figure 3D: How relative abundance is calculated? They are not quantitative data (considering the explanation in page 18-19 lines 561-577) and normalized peak area right? What this range means with respect to the concentration of these metabolites in plasma? Higher

log₂ (response) between two metabolites does not translate their real concentrations but just increase and decreased peak area. I am curious about matrix effect and recovery of analytical method and how it will affect the justification about peak area comparison. Use of peak area could be misleading considering the analytical method performance and most importantly ionization efficiency of metabolites (sometimes low abundant analyte can give sharp signal in MS due to their ionization efficiency).

Page 7, lines 167-168: why the specificity value is as low as 75%? This means that the model is struggling to find the true negative cases in any of classes that cause this problem. Please, provide ROC curves and confusion matrix for each modelled target class to understand which class confuses the ML.

Page 7, about the biomarkers discovery, few papers in the literature identified glucose metabolism, amino acids and fatty acids as marker in GC (DOI: 10.3748/wjg.v26.i20.2514, DOI:10.1021/acs.jproteome.2c00295, DOI:10.1007/s00216-020-02575-y, (also in urine samples, but interesting study: DOI:10.1021/acs.jproteome.1c00267), DOI:10.1001/jamanetworkopen.2021.14186, DOI:10.1002/bmc.1671). Authors have not detected these markers (just derivatives of glutamic acid is found by author as marker in page 7, line 155) and it should be discussed in the manuscript. Is it because of their lower importance and C-index, or is it because they can be manipulated by nutritional habit which excluded from authors ML model?

Page 7, line 181: define what is the normal range?

Page 8, line 203: as discussed above and considering the explanation in page 18-19 lines 561-577, the data are just normalized peak area and they are not relative concentration.

Page 8, lines 203-206: please provide biological explanation how these metabolites could be effected by GC and how specific they are with respect to progression of GC.

Page 8, line 221-222: considering the statistic and ML accuracy, the author concluded that the ML model is not benefited from inclusion of clinical data. Is this true in all stage of gastric cancer? Which stage suffers most when using only clinical data in ML model structure? Maybe by specifying the limitation of ML based on clinical data only, the combinatory use of clinical and metabolomics data could be understandable with respect to specific classes.

Page 9, lines 230-235: considering Fig 4g, half of high risk cases are survived and they are from which GC stage? It is important to provide confusion matrix and provide the false positive discovery ratio for GC stages with respect to low and high risk. Ideally, the low GC stage should not be present in high risk group and should all be present under low risk group? Another important issue with this model is that only 34 cases are deceased in contrast to survived cases that are 147 out of 181 samples. The population size for ML training would be biased more on the survival cases than deceased cases (see Fig4 A, B, E and G, the impression is that model is tuned on one class and it is less successful to identify deceased cases). Therefore, authors should also include small paragraph about limitation of this ML model.

Page 10, lines 259-260: to verify this, the list of metabolites, their MRM ion transitions, recovery, matrix effect, LOD and LOQ should be provided. Otherwise, no conclusion can be made on the "analytical reproducibility, validation or accuracy of quantitative analysis". These data are missing from the manuscript and authors are recommended to provide these data in SIF.

Page 11, line 277: fix the number of metabolite measured. It is different number (page 18, line 556, "...detection of 258 unique endogenous water soluble metabolites...") in method section.

Page 17, lines 501:517: is there any guidelines or information about the nutritional habit or diet of participants?

Page 17, line 518: it would be interesting to see how model reacts to cases that received treatments. How the metabolites were changed and what was the prediction of ML models for these samples.

Page 18, lines 544-559: please provide ionization mode for your instrumental analysis. Moreover, the metabolomics studies usually deposit their dataset to metabolomics workbench. Authors are encouraged to deposit their dataset in National Metabolomics Data Repository (<https://www.metabolomicsworkbench.org/>). As stated before, it is crucial to provide analytical validation data for these 258 target metabolites to assure/reproduce the results. Unfortunately, without these data it is not clear how well the analytical method/instrument operated. The information such as recovery data, matrix effect, limit of detection and quantification, MRM ion transitions and

adducts form are vital to communicate the results to readers.

Page 19, lines 570-577: as stated above abundance relate to real concentration data and comprised from quantitative analysis while considering the ionization efficiency, recovery and matrix effect. Authors, as written and presented, used normalized and batch corrected peak area data as final response factor for the metabolites. Therefore, it is suggested to avoid using metabolite abundance in this context and instead use "metabolite response factor". As mentioned before, low abundant metabolite can have high ionization efficiency and give big/sharp MS signal. This however does not affect conclusion about variation in response of the metabolites between groups as long as matrix effect and recovery data are similar and it should be fine for ML developments.

Reviewer #2:

Remarks to the Author:

In the field of metabolism, the development of early detection strategies and precise postoperative interventions for gastric cancer (GC) is urgently needed due to its impact on cancer-related mortality worldwide. However, the exploration of non-invasive biomarkers for early diagnosis and patient risk assessment was underexplored prior to this work. In Chen et al., the authors conducted a targeted metabolomics analysis of 702 plasma samples from multi-center participants to investigate the metabolic changes in GC. Their machine learning analysis revealed a 10-metabolite GC diagnostic model, which they validated in an external test set. They note their model outperformed the current analytic methods for biomarker analysis (CA72-4, CA19-9, and CEA). Specifically, their machine learning-derived prognostic model demonstrated increased performance compared to traditional models because it can effectively stratify patients into different risk groups and guide personalized interventions. Collectively, their findings provide insights into the metabolic landscape of GC and identify two distinct sets of biomarkers that enable early detection and prediction of prognosis for which the primary standard of diagnosis is the costly and invasive endoscopic exam. However, enthusiasm for novelty of their work in applying machine-learning model to metabolomics analysis is diminished by the following concerns:

- For many metabolites, optimized mass spectrometry methods are needed to validate the robustness of the metabolite signatures. For instance, for GSH/GSSG, special non-oxidizing conditions are needed, and it is unclear if the authors took into consideration the stability and different optimizations needed for determining the repeatability and reliability of the 147 metabolites identified using LC-MS by optimizing for the various metabolite profiles. These experimental limitations should be characterized and addressed experimentally, as well as in the text. Especially, as glutathione metabolism was identified as the most significantly disturbed pathway in GC and it is not clear the extent to which oxidizing conditions were limited during the initial LC-MS analyses which could confound these results.

- The authors should consider their new AI model in the context of a more detailed discussion comparing it to the current machine learning algorithms used for metabolic biomarker identification analyses. For example, Metaboanalyst and Metaboverse tools (which also have LASSO and AUROC capabilities) should be cited and compared against this model, specifically for the prognostic model. It is not clear how the proposed 10-DM and 28-PM models differ from these readily-available and commonly used models (although the authors do benchmark their model performance against "traditional methods: CA72-4, CA19-9, and CEA").

- The authors tested the effectiveness of the model in diagnosing early-stage patients computationally, but this manuscript would be greatly strengthened if evidence beyond correlation for their identified biomarker targets might be provided (though literature support, clinical evidence) to demonstrate the effectiveness of their model more robustly. Alternatively, preliminary mechanistic work linking the identified metabolic targets to early-stage progression could be use full to expand this work beyond a correlative predictive model and would increase the robustness of the findings in the manuscript.

Minor comments

- The authors state "As a result, no clinically applicable biomarkers have been discovered by using metabolomics yet", however there have been metabolic biomarkers that are used in the clinic and this statement should be adjusted and references provided.

- Consideration of the 10-DM model in parallel to the CA72-4, CA19-9, and CEA performance should be discussed to contextualize these author's findings into more generalized applicability for current clinical practices.

Reviewer #3:

Remarks to the Author:

The authors performed targeted metabolomics analysis of plasma samples from multicenter patient cohorts. They identified two distinct biomarker panels that enable early detection and prognosis prediction respectively.

The aim and main results of this study focus on the non-invasive (early) diagnosis and prognosis of gastric cancer patients. Thereby, the non-invasive early diagnosis is considered to be the most interesting and relevant. Less significant is the diagnosis of late stages and their prognosis.

However, I cannot clearly see from the presentation of results that the authors' research approach is particularly suitable for early detection. Instead, I see in the figures the delineation of all stages together, but not a specific consideration for the early stages. Therefore, the authors should do a specific statistic analysis between NGC and GC UICC-stage I (or even better IA and IB). This should be done in both training and test settings, so that it becomes clear whether the models actually specifically detect UICC IA/IB. Here I also wonder if it would not be better to create an additional model/classifier just for these early stages. I would ask the authors to generate and test such a classifier.

With regard to statistical questions, I think it is very important not to shorten the presentation of results to AUC value/ROC curves, but to always provide information on sensitivity, specificity and accuracy (or F1, precision and recall).

Further, multivariate analyses should be performed, not only but especially for UICC groups along with the models/classifiers. This will reveal whether the classifiers are an independent prognostic factor or not.

It is true that the large number of cases (approx. 700) is positive. The division into subcohorts/training cohorts/test cohorts as well as diagnosis and prognosis is rather difficult in the figures, so that one has to search for the respective cohort. I think it is important for the authors to change the illustrations so that this point can be better understood intuitively.

In Figure S5 I noticed that there is no prognostic difference between adjuvant treated and otherwise treated patients (are there also neoadjuvant treated patients?). I find this remarkable, because this could mean that the therapy would not bring any benefit. Can the authors explain this? Also, there could be a correlation between the metabolome and chemotherapy. Can the authors clarify this question?

We sincerely appreciate the reviewers for your positive, constructive, and insightful comments, which significantly help us to improve our manuscript. In the revised version of the manuscript, we provided substantial revisions, including:

- 1) We conducted a stage-specific assessment of model performance, highlighting the model's effectiveness in detecting early-stage gastric cancer (GC). Furthermore, we developed a dedicated early detection model tailored specifically for stage I patients.
- 2) We provided a more detailed demonstration of the model's applicability and compared it with models based on various machine learning algorithms.
- 3) To address the major concern regarding our mass spectrometry tools and data normalization methods, we included comprehensive validation information about the LC-MS methods.
- 4) To enhance the robustness of our findings, we added discussions about the biological relevance of the modeling metabolites to GC.

In addition, we have made extensive efforts to provide a point-by-point response (highlighted in blue) to each point raised by the editors and reviewers. We hope that these responses have addressed all the concerns/advice, and improved our manuscript substantially.

Please note that figures and tables exclusively presented in this point-by-point response letter are labeled as "Response Letter Figure/Table," while others are referenced by the same serial numbers as those in the revised manuscript.

Once again, we would like to express our sincere gratitude for your time and efforts. Thank you very much!

RESPONSE TO REVIEWERS' COMMENTS

Reviewer #1 (Remarks to the Author):

The authors have performed a targeted metabolomics study on cohorts diagnosed with gastric cancer and created machine learning-based model to enable a plasma-based non-invasive early diagnostic tool. The research question is of interest for precision medicine against gastric cancer and it can help decrease treatment plans/costs. Although, this is very well-structured manuscript with respect to utility and diagnosis capability, there are some shortcomings and improvements that should be considered before accepting the manuscript. Therefore, my feedback is “major revision”.

Response: We appreciate your time and efforts in reviewing our manuscript, along with your positive and constructive comments aimed at strengthening our paper.

1. Page 3, lines 52-54: There are few studies with large cohorts and common metabolites. Although these works, including the current study by the authors, have metabolites that could be triggered by the other diseases/cancer (for instance imbalance level of GSH:GSSG associated with lung cancers, HIV etc and it generally indicates the progression of immune dysfunction or neopterin level in prognostic model which is also seen elevated in diseases such as measles and influenza virus disease). Therefore, it is barely explained biologically how specific are the biomarkers suggested for gastric cancer by author and this claim remains still open and unanswered.

Response: We understand the concerns from the reviewer. “Biomarkers can be correlational (that is, only associated with disease) and/or functional (that is, they have an identified mechanism of action related to disease). Functional biomarkers can also be used as potential therapeutic targets.” (Vargas et al., Nat Rev Cancer 2016)¹.

For the former class, single metabolite and/or metabolite panels are found to display differential changes in tumors and could be employed as biomarkers to differentiate diseases from controls²⁻⁶. Among these metabolites, oncometabolites such as succinate, lactate, fumarate, and kynurenine have been demonstrated to be associated with various cancers^{7,8}. For example, previous studies have summarized the biomarker role of succinate in multiple tumors, including neuroendocrine tumors, prostate cancer, colorectal cancer, hepatocellular carcinoma, esophageal cancer, bladder cancer, and gastric cancer, underlying the correlation of succinate level with these cancers^{8,9}. Meanwhile, the accumulation of lactate and its function in regulating cancer cells and the metabolic microenvironment of cancer have been well-established in a variety of solid human tumors, such as colon, glioblastoma, breast, prostate, stomach, and others^{10,11}. In summary, while these metabolites may not exhibit disease-

specific alterations, they remain adequate for specific disease diagnosis. Our rigorous validation of the model performance demonstrates that our model holds significance for diagnosing gastric cancer.

As for the functional biomarker, a representative example would be D-2-hydroxyglutarate (D-2-HG), which is the product of the isocitrate dehydrogenase 1 (IDH) mutation. This biomarker is found in glioblastoma, acute myeloid leukemia, intrahepatic cholangiocarcinoma, and chondrosarcomas^{12,13}. In addition, mutations of genes involved in succinate, and fumarate metabolism were found to promote tumorigenesis in various cancer, including paraganglioma, leiomyoma and leiomyosarcoma. These findings highlight the universal role of these metabolite as biomarkers for various diseases^{10,14}. Taken together, these findings implicate that both classes of biomarkers are less likely to be specifically changed in one particular disease.

Secondly, metabolism is highly conserved among various species. It is a common observation that alterations in metabolic pathways and metabolites are frequently shared across various diseases¹⁵⁻²¹. Hence, it is less likely that a unique metabolite exclusively characterizes GC. Our research methodology primarily focuses on discerning metabolic distinctions between individuals with gastric cancer (GC) and those without (NGC), rather than the pursuit of novel metabolite discovery. This comprehensive investigation involves the monitoring of 258 metabolites spanning essential metabolic pathways, encompassing energy metabolism, carbohydrate metabolism, amino acid metabolism, and nucleotide metabolism. This thorough analysis aims to identify potential disease biomarkers, a strategy that has exhibited success in numerous prior studies^{2,22-27}.

Thirdly, it's important to note that although individual metabolites may lack specificity for gastric cancer (GC), our metabolome-dependent prediction model has undergone rigorous validation. This process consistently attests to its robust ability to accurately distinguish GC patients and predict their prognosis with high specificity, as extensively detailed in the manuscript. Furthermore, our study's findings regarding metabolic features in GC align with previous research on gastrointestinal tumors. These prior studies have independently identified differential metabolites, including succinate^{28,29}, uridine³⁰, arginine³¹, and lactate³¹. This convergence of results strengthens the evidence supporting the identification of metabolic changes associated with GC in our study.

2. Page 4, lines 68-69: better to rephrase this to: “...validated in two external subsets of test cases test set 1(same cohort) and test set2 (from different cohort)”

Response: We thank the reviewer for the constructive suggestion and have already rephrased the text as suggested on lines 75-77 of page 4.

3. Page 5, lines 86-89: it is stated in the page 18, lines 556-558 that 258 unique metabolites were measured while here the authors mentioned that 147 metabolites are measured and used. Which one is the correct number of metabolites that were measured by author's method? Also the list of metabolites, their LOQ and LODs as well as ion transitions for MRM, main adduct ions and retention time should be provided in SIF to verify analytical method validation/limitation.

Response: We appreciate the reviewer for pointing this issue out. To make it clear, we actually **monitored** 258 metabolites in our method and **detected** 147 metabolites from the samples. Specifically, we employed an LC-MS-based targeted metabolomics method that covers 258 water-soluble metabolites with important biological functions, including those involved in energy metabolism, carbohydrate metabolism, amino acid metabolism, nucleotide metabolism, and more, to analyze 702 plasma samples from both GC and NGC groups. Our approach practices the widely accepted "80% rule"³², which defines a metabolite as detectable when it is identified in at least 80% of the samples within a specific group. Following this rule, 147 metabolites were robustly detected across all the 702 samples with a small proportion of undetected metabolites (<1/5 samples) were filled with a detection baseline value, 1,000, to allow the following statistical analysis. To enhance reader comprehension, we included an extensive description in the Methods section of the revised manuscript, spanning from lines 678 to 684 of page 20.

In addition, to illustrate the robustness of our detection method, we provided a dataset of methodological dataset in **Supplementary data 1**, encompassing a list of the 258 monitored metabolites, along with pertinent information concerning their ionization modes, detection limits, and quantification limits. Furthermore, we included data regarding intra-day and inter-day precision assessments for both metabolite standards and plasma quality control samples, as well as data for 293T cells. As we are currently preparing an academic paper on our targeted metabolomics method, we regret that we are unable to provide information about ion transitions for multiple reaction monitoring (MRM), main adduct ions, and retention times currently. We have applied this method in several metabolomics studies across diverse fields, including heart regeneration (*Nakada et al., Nature 2017*)³³, haematopoietic stem cells (*Agathocleous et al., Nature 2017*)³⁴, cancer (*Piskounova et al., Nature 2015, Kim et al., Nature 2017, Huang et al., Cell Metabolism 2018 and Nie et al., Nature Communications 2021*)^{22,35-37}, viral infectious diseases (*Li et al., Science Translational Medicine 2018, Xiao et al., Nature Communications 2021 and Pang*

et al., Nature Metabolism 2021³⁸⁻⁴⁰ and embryo development (Zhao et al., Nature Metabolism 2021)⁴¹.

4. Page 5, lines 94-96: the prognostic model with 28 variables seem to suffer from few deceased cases (34 deceased against 147 survived) and heavily trained/biased on survival case. If this is the case, any of orthogonal modelling approaches or one-class modelling approach should be adopted. Otherwise, author should provide proof that the prognostic model is unbiased.

Response: We appreciate the reviewer for the constructive comment and valuable advice. We elucidated the objectives and technical details of our model to showcase its applicability. Furthermore, we included a discussion concerning the utility of the one-class model within the context of this project.

While we acknowledge that class imbalance can pose issues due to the definition of the loss function, such as in the case of binary results where maximizing accuracy can lead to problems in highly imbalanced data, this challenge can be addressed by employing scoring metrics that aren't dependent on category decision thresholds, thus mitigating prevalence-related bias. On the other hand, based on the occurrence of survival outcomes, data in survival analysis are often categorized into two distinct groups: endpoint events (such as mortality) and censoring data (endpoint events remain unobserved until the end of the study, with the only knowledge that survival times exceed the observation period). Among the 181 GC patients we collected data from for constructing the survival analysis model, 34 suffer from endpoint events during the follow-up period, while the remaining 147 patients are not observed to suffer from endpoint events during the follow-up period. Due to variations in the follow-up start times and total follow-up duration among different patients, it would be incorrect to classify the prognosis risk of these patients solely based on whether the endpoint events were observed (For example, patients who have suffered from endpoint events are classified as high-risk and others are classified as low-risk).

The model we selected to construct the prognostic model, the Random Survival Forest (RSF) model⁴², is an extension of the Random Forest (RF) model specifically designed for right-censored survival data. In Brief, RSF uses survival times as the objective variable Y, not whether endpoint events were observed. Here Y represents the survival time, defined as $Y = \min(T_0, C_0)$, where T_0 is the duration until endpoint events, C_0 represents the duration of no occurrence of endpoint events during the follow-up period. The sample data consists of (T_i, X_i, δ_i) , where T_i represents the survival time of the i -th sample, X_i represents the feature vector characterizing the i -th sample, and δ_i

indicates the survival status of the i-th sample. As shown in the **Response Letter Figure 1a-b**, the RSF model generates probability curves that describe how sample probabilities change over time with survival and cumulative hazard curves. It also calculates risk scores. Higher risk scores indicate a higher likelihood of experiencing endpoint events in the future. By setting a risk score threshold, we classify patients into high and low-risk groups. Furthermore, RSF uses the C-index to measure the performance of models built using censored data. In simple terms, when calculating the C-index, all data points are paired with each other. For each permissible pair, if the sample with a shorter survival time predicts a higher risk score, it is assigned a score of 1; otherwise, it receives a score of 0.5. In other words, samples with shorter survival times are assigned higher risk scores. When the C-index is closer to 1, it indicates better predictive performance of the model, while random prediction results in a C-index of 0.5.

As the reviewer suggested, we also adopted the one-class model approach⁴³. During the modeling, we found that this modeling method is suitable for predicting the deceased/survived status at a single time point. We determined patients deceased/survived status according to their follow-up information at the time point of three-year after patient enrollment. We used the One-Class SVM to construct a predictive classification model. Its principle is to find a hyperplane that isolates the positive samples within the dataset. New samples are classified based on whether they fall within this hyperplane; if they do, they belong to the same class, otherwise, they are considered as outliers. Since it focuses only on learning patterns from one class of samples, it mitigates the issue of training errors caused by class imbalance. We used the OneClassSVM function from the 'sklearn.svm' package in Python to build the prognostic classification model. The training and testing sets were split in a 2:1 ratio. We set the hyperparameters nu=0.2 and kernel='rbf' to train the model. In the training set, the accuracy was 0.694, precision was 0.208, and recall was 0.217. In the testing set, the accuracy was 0.716, precision was 0.1, and recall was 0.111. Both confusion matrices are presented in the **Response Letter Figure 1c-d**.

In summary, the model we constructed using RSF is a prognostic risk model, not a prognostic classification model. The objective variable Y is survival time, not whether endpoint events were observed, so there is no issue of class imbalance. Additionally, we used the C-index to assess the performance of the model built with censored data, considering both survival time and endpoint event occurrence, ensuring the accuracy of prediction errors. Furthermore, we attempted to use One Class SVM and similar modeling methods to build a prognostic classification model to predict whether endpoint events would occur at a single time point. However, the model's predictive

performance was not satisfactory. Therefore, we believe that using RSF to construct a prognostic risk model is a suitable and more effective approach for our purposes.

Response Letter Fig. 1 | a, Survival probability plot for sample data. **b,** Cumulative hazard curves for sample data. **c,** Confusion matrix of the one class SVM for training set. **d,** Confusion matrix of the one class SVM for test set.

5. Page 5, lines 102-103: the cases are overlapping in the score plot and they are barely distinguished. Please, indicate the variance explained and reinterpret/rephrase the conclusion from PCA analysis.

Response: We thank the reviewer for the question and advice. Firstly, we presented a 3D-PCA plot to illustrate the differential metabolic profiles between GC and NGC (**Response Letter Figure 2**). Secondly, we moderated the statement from the PCA analysis as the reviewer suggested on lines 110-112 of page 5. Additionally, it is worth noting that samples obtained from participants in large clinical cohorts yield more discrete data points on the PCA plot compared to those derived from cell lines or mouse models, owing to increased heterogeneity. Partial overlap in PCA plots in omics data obtained from large clinical cohort samples is a common occurrence. (see **Figure**

5A in Wolrab, D., et al., *Nat Commun* 2022, Figure 2A in Nie et al., *Nat Commun* 2021, Figure 1A in Agarwal et al., *Front Endocrinol (Lausanne)* 2022, Figure 1A in Wang et al., *Front Oncol* 2022b, Figure 2A in Shanmuganathan et al., *Front Mol Biosci* 2021, Figure 3A in Hoel et al., *JCI Insight* 2021)^{5,22,23,27,44,45}.

Response Letter Fig. 2 | 3D-PCA (Principal Component Analysis) plot of the plasma targeted metabolomics data comparing GC patients and NGC controls.

6. Pages 5 and 6, lines 101-136: the detected and selected metabolites are required to be discussed in more details from biological perspectives and how specific they are with respect to gastric cancer. It should be stated clearly how common are these metabolites between many other cancers and gastric cancer. For instance, as stated before, GSH:GSSG associated is with many other underlying conditions and is not gastric cancer specific. These metabolites could be affected by diet habit, aging or other diseases. What is missing from cohort information is about possible other underlying diseases as well as their nutritional habit. Therefore, the discussion should be reached and states biologically how relevant these metabolites are to the GC and if affected by diet habit or underlying diseases since the ML model will use them to decide about GC positive and false positive discovery rate should be addressed considering these facts.

Response: We thank the reviewer for the valuable advice and we incorporated a biological discussion of the model metabolites in response to the comment on lines 160-165 of page 7.

As previously addressed in response to Question 1, it is improbable that the detected metabolites are specific to GC. Disruptions in oncometabolites such as succinate and lactate have been discussed in relation to GC and other cancers in response to Question 1. Additionally, alterations in the model metabolite uridine were observed in both GC tumor tissues³⁰ and plasma of prostate cancer⁴⁶. Studies have reported a decrease in S-adenosylmethionine (SAM) in GC tumor tissues, which suppressed tumor proliferation and progression^{47,48}. Furthermore, the role of SAM and its synthase in tumorigenesis has been demonstrated in studies related to liver, colon, gastric, breast, pancreas, and prostate cancer⁴⁹. Another model metabolite, pyroglutamate, exhibited differential levels between tumors and controls, including gastric cancer^{50,51}, non-small cell lung cancer⁵², cervical cancer⁵¹, breast cancer⁵³, and esophagogastric junction adenocarcinoma⁵⁴. The elevation of serum neopterin level has been replicated in various cancers, including breast cancer, colorectal cancer, gastrointestinal tumors, gynecological cancer, lung cancer, hepatocellular cancer, and melanoma, all exhibiting a consistent direction of effect⁵⁵⁻⁵⁸. In summary, the alteration of single metabolites does not appear to be specific to GC. However, it is important to note that this does not impact the conclusions we have drawn or the predictive models we have developed. Our approach relies on a set of metabolites rather than a single metabolite, a strategy that has demonstrated its effectiveness in numerous prior research studies.

We thank the reviewer for raising the question regarding the potential impact of diet habits, aging, and underlying diseases on metabolites. We did not include these factors in our study design for several reasons: Firstly, our primary objective was to establish

robust prediction models that would remain unaffected by irrelevant variables. To achieve this goal, we made extensive efforts to assemble a diverse cohort to enhance the future clinical applicability of our model. Both the GC and NGC groups were randomly collected to mitigate the influence of dietary bias. Furthermore, these two groups encompassed a wide range of age groups, making them suitable for future biomarker screening programs. Secondly, neither diet habits nor information related to underlying diseases were incorporated into the machine learning modeling process. The primary focus of our model was to determine two dependent variables: GC status (GC or NGC) and survival status (survived or deceased). Specifically, we used only metabolomics data and GC status information to construct the 10-DM diagnostic model, while the 28-PM prognostic model was constructed using metabolomics data and follow-up information. Additionally, we reviewed studies examining the influence of dietary habits on metabolite changes, and found that the metabolites influenced by different diet habit including short chain fatty acids, cholesterol, phospholipids, acetate, and amino acids such as serine, glycine, aspartate, and histidine⁵⁹⁻⁶⁴. However, based on our extensive knowledge from these papers, none of the model metabolites were reported to be influenced by diet habit and related with the GC status or survival status^{62,65}. Lastly, we noted that other studies focusing on biomarker discovery based on large clinical cohorts also did not take nutritional habits into consideration^{2,26,66-69}. In summary, we contend that the omission of these factors will not compromise the applicability of our model.

7. Pages 6-7, line 138-157: please describe clearly how metabolites such as succinic acid, uridine, lactate etc are varied or affected by GC biologically. How specific these metabolites are in GC positive cases and other diseases and show limit of ML model with respect the elevated level of these metabolites and negative GC. Other confusing case is that looking at Figure 3D, most of these metabolites with exception of Uridine and lactate have wide distribution in their log2 (relative abundance) and overlaps with non-gastric cancer cases. For instance, considering the stage I, II, III with NGC in SAM metabolite, they have almost similar distribution (percentile 85%). Please, provide p-value and significance test group wise to understand clearly the Figure 3D. Show the level of significance for each pairwise comparison and provide explanations why the overlap between different stages of GC and NGC are closely related. Should another scaling or normalization method to be used?

Response: We thank the reviewer for the valuable advice. We incorporated a biological discussion of the model metabolites in **response to Question 1 and Question 6** and on lines 160-165 of page 7 in the revised manuscript.

The effect of log₂ scaling on values with large relative analytical standard deviations is problematic, as it tends to accentuate bias for metabolites with relatively lower concentrations. In contrast, the z-score utilizes the standard deviation as the scaling factor, making it more suitable for large clinical samples characterized by strong heterogeneity and large standard deviations⁷⁰. In addition, we conducted a comparative analysis of violin plots depicting response signals using three different approaches: raw normalized peak areas, z-score transformed normalized peak areas, and log₂ scale transformed normalized peak areas for the ten model metabolites (**Response Letter Figure 3**). Our investigation revealed that the use of the z-score scaling method does not significantly alter the distribution of the original data. However, we observed that log₂ scaling provides a more detailed view of the distribution but may introduce graphical modifications that could potentially mislead readers. Therefore, we have chosen to present the data in **Extended Figure 3A** using z-score transformed normalized peak areas for the ten model metabolites.

For a deeper understanding of the data presented in the previous Figure 3D (now **Supplementary Figure 3A**), we included grouped p-values in **Supplementary Data 2**. It is important to note that all ten model metabolites are differential metabolites distinguishing GC from NGC.

As explained in **response to Question 5**, samples from large clinical cohorts have higher heterogeneity and dispersion than samples from cell lines or mouse models. It is not uncommon to observe overlapping biomarker distributions among differentiating groups (see **Figure 2B** in *Sammut et al., Nature 2022*, **Figure 4B** in *Roy et al., Mol Cancer 2022*, **Extended Figure 6A-C** in *Niu et al., Nat Med 2022*, **Figure 1E** in *Oh et al., Cell Metab 2020*)⁷¹⁻⁷⁴.

Response Letter Fig. 3 | a, Violin plots of the ten metabolites in 10-DM model using the original normalized peak area. **b**, Violin plots of the ten metabolites in 10-DM model using z-score transformed normalized peak area. **c**, Violin plots of the ten metabolites in 10-DM model using \log_2 -scale transformed normalized peak area. The difference was calculated by the two-sided Kruskal-Wallis test. Black dots represent population medians.

Figure 2B in Sammut et al., 2022

Figure 4B in Roy et al., 2022

Extended Figure 6A-C in Niu et al., 2022

E microbiome: discriminatory species

Figure 1E in Oh et al., 2020

Extended Data Fig. 3 | 10-DM model metabolites distribution and performance evaluation.

a, Violin plots of the modeling metabolites using the z-score transformed normalized peak area among Cohort 1, including NGC (n=281) and GC (Stage I, n=52; Stage II, n=30; Stage III, n=53; Stage IV, n=10) plasma samples. The differences were calculated using the two-sided Kruskal-Wallis test. Black dots represent population medians. **b**, The prediction performance of the 10-DM model for distinguishing stage II/III/IV GC from NGC in test set 1 and test set 2. The dotted line represented the cut-off value of 0.50 used to separate the predicted NGC (on the left side) from GC (on the right side).

8. Page, 7, line 157 (Fig 3d) and page 26, Figure 3D: How relative abundance is calculated? They are not quantitative data (considering the explanation in page 18-19 lines 561-577) and normalized peak area right? What this range means with respect to the concentration of these metabolites in plasma? Higher log2 (response) between

two metabolites does not translate their real concentrations but just increase and decreased peak area. I am curious about matrix effect and recovery of analytical method and how it will affect the justification about peak area comparison. Use of peak area could be misleading considering the analytical method performance and most importantly ionization efficiency of metabolites (sometimes low abundant analyte can give sharp signal in MS due to their ionization efficiency).

Response: Thanks for the question from the reviewer. In response, we addressed the query through the following avenues: an elucidation of the preceding steps involved in the Figure 3D of our original manuscript, revised version of the previous Figure 3D, and evidence substantiating the efficacy of our method.

The relative abundance presented in the previous Figure 3D was calculated through the following procedures: Firstly, we applied a \$\log_2\$ transformation to the original normalized peak area. Then, we computed the mean of the original normalized peak area for each individual metabolite in the NGC group. Thirdly, we divided the original normalized peak area values of both the NGC group and the GC group at different stages by the mean value calculated in step 2. In this step, we standardized the NGC group to 1 and obtained the relative value for the GC group (relative to NGC). Consequently, the y-axis appears to have a similar range; however, this does not imply that the relative abundance of these 10 metabolites is comparable, as they were detected using a relative quantitative method. The y-axis ranges on the plot are independent of the actual concentration of these metabolites in plasma, and therefore, the relative abundance of a single metabolite can only be compared between the NGC group and different GC stages.

We agree with the reviewer that the previous data presentation method may potentially confuse readers. To avoid confusion, we present the z-score transformed original normalized peak area in **Extended Figure 3A** without employing the relative ratio calculation procedures described above.

In response to the reviewer's query concern about our methodology, we provided a dataset of methodological dataset in **Supplementary data 1**, encompassing a list of the 258 monitored metabolites, along with pertinent information concerning their ionization modes, detection limits, and quantification limits. To demonstrate the reproducibility and reliability of our analytical method, we included data regarding intra-day and inter-day precision assessments for both metabolite standards and plasma quality control samples, as well as data for 293T cells. Due to all the 258 monitored metabolites in our methodology originate from endogenous sources, the direct evaluation of their matrix effects and recovery rates within the plasma matrix presents significant challenges. To demonstrate the reproducibility of our method, we employed

our analytical method to analyze five stable isotope-labeled metabolite standards with varying physical and chemical properties, thus simulating the relevant detection process. The corresponding results are presented in the **Response Letter Table 1.**

Response Letter Table 1. Matrix effect and recovery of five stable-isotope-labeled metabolite standard

Analyte	Matrix effect (RSD, n=6)			Recovery (RSD, n=3)		
	Low	Medium	High	Low	Medium	High
Succinate- ¹³ C2	3.82%	6.52%	14.14%	2.86%	3.92%	11.75%
Methionine- ¹³ C5	14.54%	1.21%	1.28%	1.36%	1.43%	0.23%
D4-Nicotinamide	0.97%	1.76%	1.47%	1.44%	4.56%	0.94%
Arginine- ¹³ C6	13.47%	7.75%	7.10%	18.54	4.48%	1.91%
Adenosine-ribose- ¹³ C5	3.16%	0.77%	1.17%	9.36%	0.60%	2.14%

9. Page 7, lines 167-168: why the specificity value is as low as 75%? This means that the model is struggling to find the true negative cases in any of classes that cause this problem. Please, provide ROC curves and confusion matrix for each modelled target class to understand which class confuses the ML.

Response: We realized that the previous Figure 3e-f may have caused confusion for the reviewer, leading to a misunderstanding of the application of our 10-DM GC diagnostic model. Indeed, the primary function of this model is to predict and distinguish GC from NGC, rather than predicting the TNM stage to which the GC belongs. The different colored circles in Figure 3e-f were used to represent the actual disease status of the participants and to provide readers with a more intuitive understanding of the model's prediction accuracy for patients at each stage. We acknowledge that we included too much information in this panel without clear explanation. Therefore, we replaced the previous Figure 3e-f with **Figure 3d-g** and **Extended Figure 3b** to illustrate the model's performance on the entire NGC and GC cohort, and then to demonstrate the model's performance in detecting GC patients at each stage specifically.

In addition, from the specificity calculation formula, specificity=True Negatives / (True Negatives + False Positives), we noticed that the presence of false positives contributes to lower specificity in GC detection. Consequently, this may result in some individuals without gastric cancer receiving a positive diagnosis. Nevertheless, we don't think this impede the model's applicability for future applications in large-scale

screening. In the context of GC screening, a higher sensitivity holds greater clinical significance, as it facilitates the identification of more GC patients and the timely intervention. Subsequently, conducting gastroscopy on these individuals with false positive results allows for further determination of their actual GC status. This approach is considered more cost-effective, efficient, and with higher compliance than subjecting everyone to gastroscopy screening.

Furthermore, we provided ROC curves and a confusion matrix for each stage to enhance the comprehension of the model's performance on different stages (Response Letter Figure 4).

Fig. 3 | Machine learning-derived prediction model based on plasma metabolome for GC diagnosis.

a, Design of the modeling workflow. LASSO regression and random forest algorithm were adopted for feature selection and model training. The 10-DM model was validated in a test set and an external test set. **b**, The Receiver operating characteristic (ROC) curve for the diagnosis of GC patients in the test set. A 95% confidence interval was calculated based on the mean and covariance of one thousand random sampling tests. **c**, Contribution of the ten metabolites to the 10-DM model. **d-g**, The prediction performance of the 10-DM model for distinguish GC

from NGC in the test set 1 (d) and the test set 2 (e) and for distinguishing stage I GC patients from NGC in the test set 1 (f) and the test set 2 (g). The dotted line represented the cut-off value of 0.50 used to separate the predicted NGC (on the left side) from GC (on the right side).

Response Letter Fig. 4 | a, ROC curves for NGC and GC patients at each stage in test set 1 and test set 2. b, Confusion matrix for NGC and GC patients at each stage in test set 1 and test set 2.

10. Page 7, about the biomarkers discovery, few papers in the literature identified glucose metabolism, amino acids and fatty acids as marker in GC (DOI: 10.3748/wjg.v26.i20.2514, DOI:10.1021/acs.jproteome.2c00295, DOI:10.1007/s00216-020-02575-y, (also in urine samples, but interesting study: DOI:10.1021/acs.jproteome.1c00267), DOI:10.1001/jamanetworkopen.2021.14186, DOI:10.1002/bmc.1671). Authors have not detected these markers (just derivatives of glutamic acid is found by author as marker in page 7, line 155) and it should be discussed in the manuscript. Is it because of their lower importance and C-index, or is it because they can be manipulated by nutritional habit which excluded from authors ML model?

Response: We extend our appreciation to the reviewer for the question. We incorporated a discussion about previous GC markers on lines 56-61 of page 3.

We did not select the metabolites mentioned in the papers referenced by the reviewer as markers for several reasons. Firstly, as outlined in the **response to Question 1**, our targeted metabolomics method monitored only 258 metabolites and did not cover metabolites such as threonine, glutaric acid, suberic acid, 3-hydroxypropionic acid, 3-hydroxyisobutyric acid, octanoic acid, phosphoric acid, \$\alpha\$ -linolenic acid, linoleic acid, and palmitic acid, which were identified as biomarkers in the referred papers^{31,75-79}. Secondly, these referenced studies were based on serum, fecal water, urine, and tissue samples, utilizing LC-MS and NMR methods, which differ from our study that detected metabolites in plasma samples using LC-MS-based targeted metabolomics^{31,75-79}. Therefore, while we did confirm the differential presence of metabolites such as arginine, lactate, and succinate in our work, it is reasonable that we did not replicate the entire metabolic profile and markers due to differences in sample type and methodology. Thirdly, as mentioned on lines 334-336 of page 12, we considered all 147 metabolites in our analysis and used the LASSO algorithm to automatically select important features based on metabolite importance (detailed in the Method section). Throughout this process, we did not manipulate or exclude any metabolites from the potential marker panel.

Furthermore, as stated in the **responses to Questions 3 and 6**, to the best of our knowledge, we have not found any studies indicating that model metabolites are affected by diet and are related to the occurrence and progression of GC.

11. Page 7, line 181: define what is the normal range?

Response: We thank the reviewer for the question and have already include the information of the normal range on lines 202-203 of page 8. “CEA: 0-5µg/L, CA19-9: 0-27U/mL, CA72-4: 0-6.9 U/mL”

12. Page 8, line 203: as discussed above and considering the explanation in page 18-19 lines 561-577, the data are just normalized peak area and they are not relative concentration.

Response: We thank the reviewer for pointing out the problem. We rephrased the text into “relative abundance” as the data are normalized peak area on line 232 of page 8.

13. Page 8, lines 203-206: please provide biological explanation how these metabolites could be effected by GC and how specific they are with respect to progression of GC.

Response: We appreciate the reviewer for the advice and have already included the discussion on lines 236-252 of page 9.

Eleven metabolites significantly differentiated GC patient outcomes in the 28-PM model, among which elevated levels of **14:0 carnitine** and **8:0 carnitine** were associated with a worse outcome. Previous studies on GC have identified increased expression of CPT1, the rate-limiting enzyme regulating long-chain fatty acid oxidation, accelerating GC progression. The expression levels of CPT1C could also affect the outcome of GC patients. Moreover, the role of CPT1 in other cancers has also been reported, suggesting that fatty acid metabolism might play a vital role in cancer metabolic adaptation⁸⁰⁻⁸².

In addition, elevated levels of **neopterin** were indicative of a poor prognosis. Neopterin is produced by macrophages or DC cells stimulated by IFN γ , commonly regarded as one of the biomarkers for immune activation⁸³. In a single-cell transcriptomic study of GC, it was found that macrophages in tumor microenvironment play multiple roles in modulating tumor immunity⁸⁴. Furthermore, neopterin has been demonstrated in various studies to possess the potential capability for prognosis monitoring including endometrial cancer, prostate cancer, colorectal cancer, and gastric cancer⁸⁵⁻⁸⁸, which might explain the elevated plasma levels of neopterin.

GlcNAc6p is an intermediate metabolite in hexosamine pathway, a glucose metabolism pathway. Its downstream metabolite, UDP-GlcNAc, has been identified as an oncometabolite that activates the oncogenic and tumor-promoting functions of the CNC family member NRF1 by increasing its O-GlcNAcylation level⁸⁹. Stabilized NRF1 then activates the transcription of proteasome subunit genes in colon cancer,

promoting oncogenesis. Analysis of TCGA dataset indicates that the upstream and downstream enzymes of GlcNAc6p, GNPAT1 and PGM3, are upregulated in GC tumor tissues compared to normal tissue, suggesting the activation of UDP-GlcNAc6p production from GlcNAc6p (<http://gepia2.cancer-pku.cn>).

ADMA promotes the migration and invasion of gastric cancer cells through enhancing epithelial-mesenchymal transition (EMT) and regulating β -catenin expression in GC⁹⁰.

14. Page 8, line 221-222: considering the statistic and ML accuracy, the author concluded that the ML model is not benefited from inclusion of clinical data. Is this true in all stage of gastric cancer? Which stage suffers most when using only clinical data in ML model structure? Maybe by specifying the limitation of ML based on clinical data only, the combinatory use of clinical and metabolomics data could be understandable with respect to specific classes.

Response: We thank the reviewer for the construction suggestions. We conducted a further comparison of the models' application across different stages of gastric cancer to address the reviewer's inquiries.

It is not suitable to calculate the c-index (an indicator of model performance) for the test set patients in stages I and IV individually, as they all either survived or deceased. Therefore, we combined stage I and II as the early-stage group and stage III and IV as the late-stage group and assessed the model's performance on these two groups. As demonstrated in the **Extended Data Fig. 5c**, the metabolic model 28-PM exhibits greater robustness in predicting GC patients' prognosis among different stages, whereas the clinical feature model shows a poor predictive effect. The metabolic model that integrates clinical features achieves a higher prognostic prediction accuracy for early-stage patients compared to late-stage patients (C-index value 0.868 vs. 0.778). In summary, the metabolic model integrated with clinical characteristics could be applied for early-stage GC patients' prognosis prediction (with a 3% benefit compared to 28-PM model), while for late-stage GC patients, the 28-PM model performed better.

In addition, when employing clinical factors as features to construct a classification model, these factors are typically transformed into one-hot vectors. While one-hot encoding effectively preserves all information related to categorical variables, it often results in sparse data. This sparsity arises because most categories are only represented in a small subset of the samples. Sparse data, in turn, can potentially impede the performance of machine learning algorithms.

Extended Data Fig. 5 | Characteristics of the 28-PM model.

a, Weights of the 28 metabolic features in the 28-PM model. **b**, Kaplan–Meier curves for the overall survival of test set GC patients stratified by 28-PM model metabolites with a two-sided log-rank test. The patients were divided into high and low groups by the median of the metabolite abundances in GC patients. **c**, C-index comparison of models, including the 28-PM model, the 28-PM panel integrated with clinical factors, and clinical factors, for predicting the prognosis of GC patients at different stages.

15. Page 9, lines 230-235: considering Fig 4g, half of high risk cases are survived and they are from which GC stage? It is important to provide confusion matrix and provide the false positive discovery ratio for GC stages with respect to low and high risk. Ideally, the low GC stage should not be present in high risk group and should all be present under low risk group? Another important issue with this model is that only 34 cases are deceased in contrast to survived cases that are 147 out of 181 samples. The population size for ML training would be biased more on the survival cases than deceased cases (see Fig4 A, B, E and G, the impression is that model is tuned on one class and it is less successful to identify deceased cases). Therefore, authors should also include small paragraph about limitation of this ML model.

Response: We thank the reviewer for the questions and advice. First, we need to clarify that, as explained in the **response to Question 4**, our model was developed to predict the prognostic risk of GC patients. This means predicting the survival probabilities at different time points after sampling. It's important to note that the follow-up period for GC patients varies. The survival status of the patients depicted in Figure 4 is based on their survival/death status at the last follow-up time point. Additionally, the higher ratio of metastasis/recurrence also aligns with the definition of high risk, as depicted in Figure 4G. In summary, there is no conflict between a patient's survival status at the final follow-up time point and their classification into the high-risk group.

Statistical analysis revealed that among the surviving high-risk cases, 6 were in stage I, 11 were in stage II, and 23 were in stage III.

As previously explained in **response to question 4**, it would be inappropriate to categorize the prognostic risk of these patients solely based on the occurrence of the endpoint event (death), given the variations in the start of follow-up and total follow-up time among patients. Consequently, employing the confusion matrix and false positive discovery rates for low-risk and high-risk GC stages is not applicable in this context. Instead, we presented a histogram depicting the distribution of GC patients across different risk groups in terms of their stage, as illustrated in **Response Letter Figure 5**. It is noteworthy that, although the proportion of early-stage patients in the low-risk group is indeed significantly higher than that in the high-risk group, there are also early-stage GC patients predicted to be high-risk. This observation aligns with clinical observations that some stage I patients have a worse prognosis than stage III or even stage IV patients. One possible explanation for this phenomenon is that lymph node-positive stage I patients have the potential for metastasis and recurrence, which is consistent with the clinical recommendation of perioperative chemotherapy for such patients⁹¹.

We have addressed concerns related to population size in the context of our machine learning modeling in response to Question 5 and provided further discussion on lines 385-390, page 13.

Response Letter Fig. 5 | TNM stage distribution in GC patients from low-risk and high-risk group. Fisher's exact test was used to calculate the P value.

16. Page 10, lines 259-260: to verify this, the list of metabolites, their MRM ion transitions, recovery, matrix effect, LOD and LOQ should be provided. Otherwise, no conclusion can be made on the “analytical reproducibility, validation or accuracy of quantitative analysis”. These data are missing from the manuscript and authors are recommended to provide these data in SIF.

Response: We thank the reviewer for the questions and have explained in the response to Question 3.

17. Page 11, line 277: fix the number of metabolite measured. It is different number (page 18, line 556, “... detection of 258 unique endogenous water soluble metabolites...”) in method section.

Response: We thank the reviewer for the questions and have explained in the response to Question 3.

18. Page 17, lines 501:517: is there any guidelines or information about the nutritional habit or diet of participants?

Response: Thanks for the reviewer's question, we explained in the response to Question 6.

19. Page 17, line 518: it would be interesting to see how model reacts to cases that

received treatments. How the metabolites were changed and what was the prediction of ML models for these samples.

Response: We thank the reviewer for the question and advice. We applied the 28-PM model to predict the prognostic risk of 23 cases of GC patients who received neoadjuvant chemotherapy before plasma sampling. The C-index of the model performance on this GC group is 0.5, indicating that the 28-PM model is not superior to random prediction (**Response Letter Figure 6a**).

We conducted a metabolome comparison between the GC patients who received neoadjuvant chemotherapy and NGC. The findings of the differential metabolite analysis are presented in **Response Letter Figure 6b**. Furthermore, we compared the differential metabolites of neoadjuvant chemotherapy GC/NGC and GC/NGC, highlighting the overlapping differential metabolites in yellow.

Response Letter Fig. 6 | a, Prognostic prediction of the GC patients who received neoadjuvant chemotherapy using the 28-PM model. The dotted line, drawn at the cut-off value of 2.1, divides the patients into high- and low-risk groups. Green circles and gray circles represent survived and deceased respectively. **b**, The differential metabolites in GC patients who received

neoadjuvant chemotherapy versus NGC. Two-sided Wilcoxon rank-sum tests followed by Benjamini-Hochberg (BH) multiple comparison test with false discovery rate (FDR) < 0.05 and fold change (FC) > 1.25 or <0.8. Red circles and blue circles represent up-regulated and down-regulated metabolites in GC patients who received neoadjuvant chemotherapy respectively. The overlapping differential metabolites between GC/NGC and GC patients who received neoadjuvant chemotherapy/NGC are highlighted in yellow.

20. Page 18, lines 544-559: please provide ionization mode for your instrumental analysis. Moreover, the metabolomics studies usually deposit their dataset to metabolomics workbench. Authors are encouraged to deposit their dataset in National Metabolomics Data Repository (<https://www.metabolomicsworkbench.org/>). As stated before, it is crucial to provide analytical validation data for these 258 target metabolites to assure/reproduce the results. Unfortunately, without these data it is not clear how well the analytical method/instrument operated. The information such as recovery data, matrix effect, limit of detection and quantification, MRM ion transitions and adducts form are vital to communicate the results to readers.

Response: We appreciate the reviewer's valuable feedback.

We regret that we are currently unable to provide information about ion transitions for multiple reaction monitoring (MRM), main adduct ions, and retention times because we are in the process of preparing an academic paper on our targeted metabolomics method. In response to the reviewer's suggestions, we provided a dataset of methodological data encompass a list of 258 monitored metabolites, detailing their ionization modes, detection limits, quantification limits and included the raw mass dataset in the **Supplementary Data 1**.

Furthermore, to demonstrate the reliability of our methodology, we included data pertaining to intra-day and inter-day precision assessments for both metabolite standards and samples of plasma quality control and 293T cells. However, it is important to note that all 258 monitored metabolites in our methodology originate from endogenous sources. Consequently, the direct evaluation of their matrix effects and recovery rates within the plasma matrix presents significant challenges. Therefore, we employed our analytical method to analyze five stable isotope-labeled metabolite standards with varying physical and chemical properties, thus simulating the relevant detection process. The corresponding results are presented in the **Response Letter Table 1** (in the **response to Question 3 and Question 8**).

21. Page 19, lines 570-577: as stated above abundance relate to real concentration data and comprised from quantitative analysis while considering the ionization efficiency, recovery and matrix effect. Authors, as written and presented, used normalized and batch corrected peak area data as final response factor for the metabolites. Therefore, it is suggested to avoid using metabolite abundance in this context and instead use “metabolite response factor”. As mentioned before, low abundant metabolite can have high ionization efficiency and give big/sharp MS signal. This however does not affect conclusion about variation in response of the metabolites between groups as long as matrix effect and recovery data are similar and it should be fine for ML developments.

Response: We appreciate the reviewers' valuable suggestions. In response to these suggestions, we made modifications to the content of the revised manuscript. Specifically, we replaced “metabolite abundance” with “relative abundance”.

As mentioned by the reviewer, we employed a targeted metabolomics analysis method based on LC-MS for the purpose of conducting relative quantification analysis on plasma samples. Subsequently, we performed batch effect removal and data normalization on the raw data. Consequently, our primary focus was directed towards comparing the differences in relative abundance of specific metabolites between the GC and non-GC groups. It is worth noting that the relative abundance of different metabolites within a specific sample cannot be compared directly due to variations in their distinct ionization efficiencies, matrix effects, and recovery rates. Therefore, to prevent readers from erroneously assuming that different metabolites within the same sample are inherently comparable, we replaced the term 'metabolite abundance' as it appeared in the original text.

For the purpose of enhancing reader comprehension, we referred to relevant representative metabolomics literature. It was found that the “relative abundance” was commonly used⁹²⁻⁹⁴ to compare the metabolic difference between different groups.

As a result, we opted to replace “metabolite abundance” with “relative abundance”.

Reviewer #2 (Remarks to the Author):

In the field of metabolism, the development of early detection strategies and precise postoperative interventions for gastric cancer (GC) is urgently needed due to its impact on cancer-related mortality worldwide. However, the exploration of non-invasive biomarkers for early diagnosis and patient risk assessment was underexplored prior to this work. In Chen et al., the authors conducted a targeted metabolomics analysis of 702 plasma samples from multi-center participants to investigate the metabolic changes in GC. Their machine learning analysis revealed a 10-metabolite GC diagnostic model, which they validated in an external test set. They note their model outperformed the current analytic methods for biomarker analysis (CA72-4, CA19-9, and CEA). Specifically, their machine learning-derived prognostic model demonstrated increased performance compared to traditional models because it can effectively stratify patients into different risk groups and guide personalized interventions. Collectively, their findings provide insights into the metabolic landscape of GC and identify two distinct sets of biomarkers that enable early detection and prediction of prognosis for which the primary standard of diagnosis is the costly and invasive endoscopic exam.

Response: We highly appreciate your time and efforts in reviewing our manuscript and your insightful comments and suggestions for our work. We provided new data and information below and in the revised manuscript to address all your concerns about the important technical points.

1. For many metabolites, optimized mass spectrometry methods are needed to validate the robustness of the metabolite signatures. For instance, for GSH/GSSG, special non-oxidizing conditions are needed, and it is unclear if the authors took into consideration the stability and different optimizations needed for determining the repeatability and reliability of the 147 metabolites identified using LC-MS by optimizing for the various metabolite profiles. These experimental limitations should be characterized and addressed experimentally, as well as in the text. Especially, as glutathione metabolism was identified as the most significantly disturbed pathway in GC and it is not clear the extent to which oxidizing conditions were limited during the initial LC-MS analyses which could confound these results.

Response: We thank the reviewer for the constructive feedbacks.

Our mass spectrometry methods were rigorously optimized and subjected to validation to ensure the robustness, repeatability, and reliability of the 147 metabolites. This will be presented comprehensively through three key aspects: a detailed exposition of our methodology, demonstration of method reliability within the supplementary information, and a thorough discussion pertaining to the quantification of GSH/GSSG.

Firstly, our targeted metabolomics method was established on LC-QQQ-MS/MS system using MRM mode. The parent/product ions of MRM transitions of each metabolite were optimized by injecting analytical standards to the MS spectrometer. And then the retention time of each metabolite was determined by injecting analytical standards to the LC-MS system. Finally, 258 metabolites of common and important metabolic pathways (energy metabolism, carbohydrate metabolism, amino acid metabolism, nucleotide metabolism, etc) were included in this method. We have applied this method in several metabolomics studies across diverse fields, including heart regeneration (*Nakada et al., Nature 2017*)³³, haematopoietic stem cells (*Agathocleous et al., Nature 2017*)³⁴, cancer (*Piskounova et al., Nature 2015, Kim et al., Nature 2017, Huang et al., Cell Metabolism 2018 and Nie et al., Nature Communications 2021*)^{22,35-37}, viral infectious diseases (*Li et al., Science Translational Medicine 2018, Xiao et al., Nature Communications 2021 and Pang et al., Nature Metabolism 2021*)³⁸⁻⁴⁰ and embryo development (*Zhao et al., Nature Metabolism 2021*)⁴¹.

Secondly, to demonstrate the reliability of our methodology, we included data pertaining to intra-day and inter-day precision assessments for both metabolite standards and samples of plasma quality control and 293T cells in the **Supplementary data 1**, as well as the methodological data including LOD, LOQ and so on.

Thirdly, for the quantification of GSH/GSSH in the metabolomics experiments, Lu et al.,⁹⁵ showed that careful analysis is needed in the absolute quantification of redox-related metabolites, and extraction with 40:40:20 ACN: MeOH: H₂O with 0.1M formic acid could decrease the interconversion among the reduced forms and the oxidized forms, while drying samples will lead to a decrease in GSH and an increase in GSSG. In our case, a relative quantification analysis was performed by comparing the relative level of GSH/GSSG among the plasma samples. All the samples were treated in the same way, so the parallel and unbiased operation will not affect the trend of the relative difference between the two groups. Also, other papers used similar methods to measure the relative level of GSH, GSSG or GSH/GSSG among different groups⁹⁶⁻⁹⁸. In our previous work, Elena et al., showed that the GSH/GSSG ratio was always significantly higher in melanoma cells from subcutaneous tumors compared to circulating cells or metastatic nodules no matter whether metabolites were extracted with or without 0.1% formic acid³⁵ (**Response Letter Table 2**).

Response Letter Table 2 (From Nature 527, 186-191 (2015).)

	Experiment	Figure Panel	GSH/GSSG in SQ Tumour	GSH/GSSG in metastasizing cells	p-value
i	Tumour GSH/GSSG measurement with FA	Figure 1a	270	140	0.0046
ii	Tumour Metabolomics #1	Extended Figure 4a	170	20	0.0006
iii	Tumour Metabolomics #2	Extended Figure 4b	26	8.1	0.0031
iv	Circulating melanoma cell metabolomics	Figure 1b	19	4.5	0.0027
v	Sorted Cell Metabolomics with FA	Extended Figure 4c	44	38	0.0186

We also performed more experiments to compare the level of GSH, GSSG, GSH/GSSG in samples with and without Speed Vac drying. The results showed that the drying step indeed led to a decrease in GSH and an increase in GSSG which was consistent with the reviewer’s insight. However, the relative level of GSH, GSSG, GSH/GSSG between two groups (a Reduced Sample group made by a standard mix solution of 5µM GSH and 1µM GSSG, and an Oxidized Sample group made by a standard mix solution of 2µM GSH and 1µM GSSG) did not change the trend (Response Letter Figure 7), which was consistent with our previous study.

Response Letter Fig. 7 | GSH/GSSG detection of reduced and oxidized sample.

2. The authors should consider their new AI model in the context of a more detailed discussion comparing it to the current machine learning algorithms used for metabolic biomarker identification analyses. For example, Metaboanalyst and Metaboverse tools (which also have LASSO and AUROC capabilities) should be cited and compared against this model, specifically for the prognostic model. It is not clear how the proposed 10-DM and 28-PM models differ from these readily-available and commonly used models (although the authors do benchmark their model performance against “traditional methods: CA72-4, CA19-9, and CEA.”).

Response: We truly appreciate the reviewer for the constructive comment and great advice. We conducted a comparative analysis of our two models with readily-available and widely used AI models, and we will discuss the results as follows:

Firstly, we performed the performance comparison of the 10-DM model with Metaboanalyst. Specifically, we input the training and testing dataset into the Metaboanalyst⁹⁹ biomarker analysis module to build diagnostic models. In the variable selection step, we selected the top ten metabolites based on Lasso frequency and AUC rank. These metabolites include GSSG, succinate, uridine, 2-PG-3-PG, lactate, uracil, pyroglutamate, SAH, SAM, and R5P. Subsequently, we constructed diagnostic models using various algorithms, including linear SVM, PLS-DA, Random Forests, and Logistic Regression, and compared them with the 10-DM model (model performance is shown in **Extended Data Fig. 4c**). Except for the PLS-DA model, all other models performed well (AUC > 0.8). However, whether in the training set, testing set, or an additional testing set, the 10-DM model consistently outperformed them. Therefore, compared to the commonly used models currently available, the 10-DM model demonstrates superior predictive performance. We involved this comparison in the text on lines 210-214 of page 8.

For prognostic models, it's worth noting that machine learning algorithms available in both Metaboanalyst and Metaboverse¹⁰⁰ are typically designed for classification tasks. However, based on the occurrence of survival outcomes, data in survival analysis are often categorized into two distinct groups: endpoint events (such as death) and censoring data (endpoint events remain unobserved until the end of the study, with the only knowledge that survival times exceed the observation period). Among the 181 gastric cancer patients we collected data from for constructing the survival analysis model, 34 suffer from endpoint events during the follow-up period, while the remaining 147 patients are not observed to suffer from endpoint events during the follow-up period. Due to variations in the follow-up start times and total survival times among different patients, it would be incorrect to classify the prognosis risk of these patients solely based on whether the endpoint events were observed (For example, patients who have suffered from endpoint events are classified as high-risk and others are classified as low-risk). The model we selected to construct the prognostic model, the Random Survival Forest (RSF) model, is an extension of the Random Forest (RF) model specifically designed for right-censored survival data. In brief, RSF uses survival times as the objective variable Y, not whether endpoint events were observed. Here Y represents the survival time, defined as $Y = \min(T_0, C_0)$, where T_0 is the duration until endpoint events, C_0 represents the duration of no occurrence of endpoint events during the follow-up period. The RSF model generates probability curves that describe how sample probabilities change over time with survival and cumulative hazard curves. It also calculates risk scores. Higher risk scores indicate a higher likelihood of experiencing endpoint events in the future. By setting a risk score threshold, we classify patients into high and low-risk groups.

Certainly, we can set a specific time point (for example, at the three-year mark after patient enrollment) and use the deceased/survived status of patients at that time point as the prediction target, effectively converting survival analysis into a classification task. However, in doing so, we would encounter the issue of class imbalance (34 deceased against 147 survived). Class imbalance is really an issue as the loss function is defined (e.g., naively training on maximizing accuracy for binary outcomes will lead to problems in highly imbalanced data). Unfortunately, it's important to note that the machine learning algorithms available in Metaboanalyst and Metaboverse may not effectively address class imbalance. Attempting to use these algorithms without addressing class imbalance could result in model bias, where the model's predictions tend to favor the majority class. We used the One-Class SVM to construct a predictive classification model. Its principle is to find a hyperplane that isolates the positive samples within the dataset. New samples are classified based on whether they fall within this hyperplane; if they do, they belong to the same class, otherwise, they are considered as outliers. Since it focuses only on learning patterns from one class of samples, it mitigates the issue of training errors caused by class imbalance. We used the OneClassSVM function from the sklearn.svm package in Python to build the prognostic classification model. The training and testing sets were split in a 2:1 ratio. We set the hyperparameters nu=0.2 and kernel='rbf' to train the model. In the training set, the accuracy was 0.694, precision was 0.208, and recall was 0.217. The confusion matrix is shown below. In the testing set, the accuracy was 0.716, precision was 0.1, and recall was 0.111. The confusion matrix is previously provided in **Response Letter Figure. 1**. The One-Class SVM model's predictive performance was not satisfactory. Therefore, we believe that using RSF to construct a prognostic risk model is a suitable and more effective approach for our purposes.

Extended Data Fig. 4 | Diagnostic accuracy comparison of the 10-DM model and clinical markers. **a**, Diagnostic prediction of the test set patients using the 10-DM model and clinical markers respectively. The training set, test set 1 and 2 GC patients were colored in blue, green, and purple respectively. The blue dotted line represents the log₂ cutoff value of each marker, while the red dotted line represents the cutoff value of the 10-DM model. **b**, Comparison of different markers and models' detection sensitivity in predicting GC patients. CEA, carcinoembryonic antigen; CA19-9, carbohydrate antigen 19-9; CA724, carbohydrate antigen

724. c, The AUROC curves for the 10-DM model and various machine learning models constructed using Metaboanalyst are depicted for the discovery set, test sets 1 and 2.

Response Letter Fig. 1 | a, Survival probability plot for sample data. b, Cumulative hazard curves for sample data. c, Confusion matrix of the one class SVM for training set. d, Confusion matrix of the one class SVM for test set.

3. The authors tested the effectiveness of the model in diagnosing early-stage patients computationally, but this manuscript would be greatly strengthened if evidence beyond correlation for their identified biomarker targets might be provided (though literature support, clinical evidence) to demonstrate the effectiveness of their model more robustly. Alternatively, preliminary mechanistic work linking the identified metabolic targets to early-stage progression could be use full to expand this work beyond a correlative predictive model and would increase the robustness of the findings in the manuscript.

Response: We thank the reviewer for the great suggestions. We made our greatest efforts to collecting mechanistic studies that demonstrate the role of identified metabolites in promoting tumorigenesis and disease progression in GC and other cancers, and we included the discussion on lines 161-165 of page 7.

Among the ten model metabolites, we found that succinate, lactate, SAM, and serotonin are associated with the early-stage progression of GC. **Succinate** is identified as an oncometabolite that acts as an oncogenic signaling molecule involved in metabolic, epigenetic alterations, and tumorigenesis⁹.

Succinate dehydrogenase (SDH) has been identified as a tumor suppressor. When it is mutated or deficient, it impedes the flow of metabolites, leading to an accumulation of succinate^{101,102}. **Succinate** acts as an antagonist to α -KG, inhibiting α -KG-dependent dioxygenases, including the JMJD family of KDMs and the TET family of 5mC hydroxylases¹⁰³. Furthermore, SDH mutation or deficiency results in widespread hypermethylation in gastrointestinal stromal tumors (GISTs). This hypermethylation disrupts CTCF insulators, allowing super-enhancers to come into contact with and activate the oncogene FGF4 both in vitro and in vivo¹⁰⁴. Furthermore, circulating **lactate** has been recognized as a primary carbon source for tricarboxylic acid (TCA) cycle and energy production in most tissues and tumors¹⁰⁵. Lactate and succinate were found to be continuously up-regulated in epithelium, serrated lesion and tumor tissues of GC patients, indicating their role in tumorigenesis and progression¹⁰⁶. **Serotonin** is a critical signaling molecule involved in gastrointestinal secretion, sensation, and peristalsis. LGR5-positive stem cells have been identified as the primary source of GC cells, and the carcinogenic potential of LGR5-positive cells has been demonstrated in animal models. In gastric mucosa tissue, serotonin and LGR5 were found to co-localize, and the expression level of TPH1, a key enzyme in serotonin synthesis, was associated with LGR5. Additionally, the positive rate of serotonin increases progressively in patients with non-atrophic gastritis, intestinal metaplasia, and gastric cancer, suggesting an association between serotonin and GC tumorigenesis¹⁰⁷.

In addition to succinate, lactate, and serotonin^{10,14}, previous research also reported associations between uridine, pyroglutamate, SAM and GlcNAc6p with the early-stage progression of other cancers. Dietary **uridine** supplementation has been shown to reduce intestinal tumor formation in the Apc Min/+ mouse model¹⁰⁸. Additionally, uridine has been identified as a biomarker for malignant transformation in cases of OELP oral erosive lichen planus¹⁰⁹. **Pyroglutamate** is a metabolite involved in the glutathione metabolism, and it has been associated with increased DNA damage and dysregulation of the γ -glutamyl cycle, promoting the transformation of ulcerative colitis into colorectal cancer. It has been identified as a biomarker for pancreatic cancer^{110,111}. Furthermore, the enzyme responsible for metabolizing pyroglutamate, 5-Oxo-L-prolinase, is downregulated in gastric tumor tissues, indicating a link between dysregulated glutathione metabolism and tumorigenesis¹¹². **SAM** and its synthase are associated with increased oxidative stress, progenitor cell expansion, and genomic instability, thereby regulating tumorigenesis in liver, colon, gastric, breast, pancreas,

and prostate cancer. It has shown effectiveness as a chemopreventive agent in rodent models of hepatocellular carcinoma (HCC) in several studies. SAM's chemopreventive properties are attributed to its ability to prevent the hypomethylation of proto-oncogene promoters in these models, leading to the induction of gene expression⁴⁹. **GlcNAc6p** is an intermediate metabolite in hexosamine pathway, a glucose metabolism pathway. Its downstream metabolite, UDP-GlcNAc, has been identified as an oncometabolite that activates the oncogenic and tumor-promoting functions of the CNC family member NRF1 by increasing its O-GlcNAcylation level⁸⁹. Stabilized NRF1 then activates the transcription of proteasome subunit genes in colon cancer, promoting oncogenesis. Analysis of TCGA dataset indicates that the upstream and downstream enzymes of GlcNAc6p, GNPAT1 and PGM3, are upregulated in GC tumor tissues compared to normal tissue, suggesting the activation of UDP-GlcNAc6p production from GlcNAc6p (<http://gepia2.cancer-pku.cn>).

4. The authors state “As a result, no clinically applicable biomarkers have been discovered by using metabolomics yet”, however there have been metabolic biomarkers that are used in the clinic and this statement should be adjusted and references provided.

Response: We thank the reviewer for the constructive suggestion. We modified the statement on lines 56-61 of page 3 and provided references.

5. Consideration of the 10-DM model in parallel to the CA72-4, CA19-9, and CEA performance should be discussed to contextualize these author’s findings into more generalized applicability for current clinical practices.

Response: We appreciate the reviewer for the valuable advice and have included the discussion on lines 206-209 of page 8. As demonstrated in **Extended Figure 4b (provided in the response to Question 2)**, the combination of the 3-biomarker panel and the 10-DM model further enhances the sensitivity of GC patient detection.

Reviewer #3 (Remarks to the Author):

The authors performed targeted metabolomics analysis of plasma samples from multicenter patient cohorts. They identified two distinct biomarker panels that enable early detection and prognosis prediction respectively.

The aim and main results of this study focus on the non-invasive (early) diagnosis and prognosis of gastric cancer patients. Thereby, the non-invasive early diagnosis is considered to be the most interesting and relevant. Less significant is the diagnosis of late stages and their prognosis.

Response: We appreciate your time and efforts in reviewing our manuscript and your positive and constructive comments to strengthen our paper.

1. However, I cannot clearly see from the presentation of results that the authors' research approach is particularly suitable for early detection. Instead, I see in the figures the delineation of all stages together, but not a specific consideration for the early stages. Therefore, the authors should do a specific statistic analysis between NGC and GC UICC-stage I (or even better IA and IB). This should be done in both training and test settings, so that it becomes clear whether the models actually specifically detect UICC IA/IB. Here I also wonder if it would not be better to create an additional model/classifier just for these early stages. I would ask the authors to generate and test such a classifier.

Response: We thank the reviewer for the question and advice.

Firstly, we replaced the previous Figure 3e-f with **Figure 3d-g** to illustrate the 10-DM model's performance on the entire NGC and GC cohort, and then to demonstrate the model's performance in detecting GC patients at stage I specifically.

Secondly, as the reviewer suggested, we constructed an additional model for the detection of stage I GC patients and achieved great model performance (**Response Letter Figure 8**). However, the accuracy of this model in predicting stage I GC patients is comparable to that of the 10-DM model, but it involves significantly fewer candidates in the modeling process (333 cases compared with 426 cases in the 10-DM model). Therefore, we conclude that the 10-DM model may be more robust and suitable for detecting GC patients.

Fig. 3 | Machine learning-derived prediction model based on plasma metabolome for GC diagnosis.

a. Design of the modeling workflow. LASSO regression and random forest algorithm were adopted for feature selection and model training. The 10-DM model was validated in a test set and an external test set. **b.** The Receiver operating characteristic (ROC) curve for the diagnosis of GC patients in the test set. A 95% confidence interval was calculated based on the mean and covariance of one thousand random sampling tests. **c.** Contribution of the ten metabolites to the 10-DM model. **d-g.** The prediction performance of the 10-DM model for distinguish GC

from NGC in the test set 1 (d) and the test set 2 (f) and for distinguishing stage I GC patients from NGC in the test set 1 (e) and the test set 2 (g). The dotted line represented the cut-off value of 0.50 used to separate the predicted NGC (on the left side) from GC (on the right side).

Response Letter Fig. 8 | a, The Receiver operating characteristic (ROC) curve for diagnosing stage I GC patients is depicted. A 95% confidence interval was calculated based on the mean and covariance of one thousand random sampling tests. **b,** Prediction of the stage I GC patients and NGC using the new stage I specific model. The dotted line represented the cut-off value of 0.50, separating the predicted NGC (on the left side) and GC (on the right side) Green circles and gray circles represent survived and deceased in the test set respectively.

2. With regard to statistical questions, I think it is very important not to shorten the presentation of results to AUC value/ROC curves, but to always provide information on sensitivity, specificity and accuracy (or F1, precision and recall).

Response: We thank the reviewer for the constructive suggestion. We provided and highlighted the complete information of sensitivity, specificity, and accuracy on lines 177-190 of page 7.

3. Further, multivariate analyses should be performed, not only but especially for UICC groups along with the models/classifiers. This will reveal whether the classifiers are an independent prognostic factor or not.

Response: We appreciate the reviewer for highlighting this problem and have already included the results of multivariate analyses in **Table 1** to demonstrate that the 28-PM classifier is an independent prognostic factor in GC. The related text is on lines 290-293 of page 10.

Table 1 Multivariate Cox regression of GC patients' prognosis in cohort 3

Characteristics	Classification	P value	Hazard ratio	95% CI
TNM staging	I	0.010	Reference	
	II		1.19	0.07 - 20.88
	III		4.02	0.24 - 66.59
	IV		12.86	0.7 - 237.08
Macroscopic appearance	Borrmann I	0.789	Reference	
	Borrmann II		3.69	0.47 - 28.78
	Borrmann III		3.35	0.43 - 25.97
	Borrmann IV		3.45	0.41 - 29.25
	EGC		1.96	0.06 - 60.26
Vascular tumor embolus	No	0.463	Reference	
	Yes		1.38	0.59 - 3.24
Metabolic risk factor*	Low	<0.001	Reference	
	High		7.53	2.83 - 20.04

* The classifier was derived from 28-PM model cutoff value, represents as a high-risk group and a low-risk group for the GC patients.
CI, Confidence interval; EGC, Early gastric cancer.

4. It is true that the large number of cases (approx. 700) is positive. The division into subcohorts/training cohorts/test cohorts as well as diagnosis and prognosis is rather difficult in the figures, so that one has to search for the respective cohort. I think it is important for the authors to change the illustrations so that this point can be better understood intuitively.

Response: We appreciate the reviewer for the great advice. To clarify the relative cohorts involved in the two models, we changed the illustration of the overview framework in the Figure 1 and specified the information in the figure legend (lines 806-818, pages 25-26).

Fig. 1 | Schematic overview of the study.

Overview of the study design. The illustration was created with BioRender.com. A total of 702 individuals were included in the study, and their plasma samples underwent targeted metabolomics analysis. The metabolic profiles of gastric cancer (GC) patients and non-GC controls (NGC) in Cohort 1 (n=426) were compared to depict the metabolic reprogramming in GC. Using the metabolomics data from Cohort 1 and machine learning techniques, a diagnostic model for GC (10-DM model) was created and validated. This model was further verified in the

test set 2 (Cohort 2, n=95). Metabolomics data from Cohort 3 (n=181) patients and their clinical features were analyzed using a machine learning algorithm to develop a prognostic model (28-PM model). Performance of these two models were benchmarked against clinically used biomarkers/clinical features. Different colored triangles in the figure represent various participant groups used for model construction, validation, and comparison processes.

5. In Figure S5 I noticed that there is no prognostic difference between adjuvant treated and otherwise treated patients (are there also neoadjuvant treated patients?). I find this remarkable, because this could mean that the therapy would not bring any benefit. Can the authors explain this? Also, there could be a correlation between the metabolome and chemotherapy. Can the authors clarify this question?

Response: We thank the reviewer for raising this intriguing question. The 28-PM model does not include GC patients received neoadjuvant chemotherapy.

We do not think that this result implies that therapy would not bring any benefit, for the following reasons: Firstly, in clinical practice, it's not suitable to require all GC patients with different TNM stages to either receive or forgo chemotherapy. Considering our primary objective of developing a widely applicable model for prognostic risk prediction in GC patients, we intentionally enrolled GC patients at various TNM stages who were undergoing standard postoperative treatment as advised by their physicians. We have constructed a Sankey diagram to depict the distribution of patient stages, whether they received adjuvant chemotherapy, their survival status, and metastasis/recurrence status at the last follow-up timepoint for your reference (**Response Letter Figure 9a**). Concurrently, proportion plots of GC patients at different stages were generated to illustrate the correlation between survival status or metastasis/recurrence status and whether they received adjuvant chemotherapy (**Response Letter Figure 9b**). The data suggests that the percentage of survivors among stage II/III GC patients who received chemotherapy was higher than those who did not, although the difference was not statistically significant. Furthermore, due to the limited cohort size and follow-up duration, any potential benefits of adjuvant therapy may not have been identified in our analysis. Additionally, upon reviewing the existing literature, we found previous studies that have demonstrated the benefits of adjuvant chemotherapy, as evidenced by a higher percentage of 3-year disease-free survival (DFS) rates and longer overall survival¹¹³⁻¹¹⁵.

To investigate the correlation between the metabolome and chemotherapy, it would be necessary to collect baseline plasma samples from patients before chemotherapy and the treatment response information after chemotherapy¹¹⁶. However, in the current study, we faced limitations as all GC patients who received chemotherapy did not develop metastasis after two cycles of treatment, making it impossible to evaluate chemotherapy response based on CT scan results. Therefore, the analysis of the

metabolome's correlation with chemotherapy outcomes was not feasible with our available sample. In the future, we intend to obtain baseline plasma samples from GC patients before they receive neoadjuvant chemotherapy and collect information on treatment response at the time of surgery to clarify the relationship between the metabolome and chemotherapy.

Response Letter Fig. 9 | a, Sankey plot showing the distribution of GC patient stage, whether received adjuvant chemotherapy, survival status, and metastasis/recurrence status at the last follow-up time point. b, Proportion plots of survival status and whether they received chemotherapy of GC patients at different stages.

Reference

- 1 Vargas, A. J. & Harris, C. C. Biomarker development in the precision medicine era: lung cancer as a case study. *Nat Rev Cancer* **16**, 525–537, doi:10.1038/nrc.2016.56 (2016).
- 2 Luo, P. *et al.* A Large-scale, multicenter serum metabolite biomarker identification study for the early detection of hepatocellular carcinoma. *Hepatology* **67**, 662–675, doi:10.1002/hep.29561 (2018).
- 3 Mahajan, U. M. *et al.* Independent validation and assay standardization of improved metabolic biomarker signature to differentiate pancreatic ductal adenocarcinoma from chronic pancreatitis. *Gastroenterology*, doi:10.1053/j.gastro.2022.07.047 (2022).
- 4 Kachroo, P. *et al.* Metabolomic profiling reveals extensive adrenal suppression due to inhaled corticosteroid therapy in asthma. *Nat Med* **28**, 814–822, doi:10.1038/s41591-022-01714-5 (2022).
- 5 Wolrab, D. *et al.* Lipidomic profiling of human serum enables detection of pancreatic cancer. *Nat Commun* **13**, 124, doi:10.1038/s41467-021-27765-9 (2022).
- 6 Abdrabou, W. *et al.* Metabolome modulation of the host adaptive immunity in human malaria. *Nat Metab* **3**, 1001–1016, doi:10.1038/s42255-021-00404-9 (2021).
- 7 Rinschen, M. M., Ivanisevic, J., Giera, M. & Siuzdak, G. Identification of bioactive metabolites using activity metabolomics. *Nat Rev Mol Cell Biol* **20**, 353–367, doi:10.1038/s41580-019-0108-4 (2019).
- 8 Yang, M., Soga, T. & Pollard, P. J. Oncometabolites: linking altered metabolism with cancer. *J Clin Invest* **123**, 3652–3658, doi:10.1172/jci67228 (2013).
- 9 Dalla Pozza, E. *et al.* Regulation of succinate dehydrogenase and role of succinate in cancer. *Semin Cell Dev Biol* **98**, 4–14, doi:10.1016/j.semcdb.2019.04.013 (2020).
- 10 Baltazar, F., Afonso, J., Costa, M. & Granja, S. Lactate Beyond a Waste Metabolite: Metabolic Affairs and Signaling in Malignancy. *Front Oncol* **10**, 231, doi:10.3389/fonc.2020.00231 (2020).
- 11 Mortazavi Farsani, S. S. & Verma, V. Lactate mediated metabolic crosstalk between cancer and immune cells and its therapeutic implications. *Front Oncol* **13**, 1175532, doi:10.3389/fonc.2023.1175532 (2023).
- 12 Dang, L. & Su, S. M. Isocitrate Dehydrogenase Mutation and (R)-2-Hydroxyglutarate: From Basic Discovery to Therapeutics Development. *Annu Rev Biochem* **86**, 305–331, doi:10.1146/annurev-biochem-061516-044732 (2017).
- 13 Choi, C. *et al.* 2-hydroxyglutarate detection by magnetic resonance spectroscopy in IDH-mutated patients with gliomas. *Nat Med* **18**, 624–629, doi:10.1038/nm.2682 (2012).
- 14 Thompson, C. B. Metabolic enzymes as oncogenes or tumor suppressors. *N Engl J Med* **360**, 813–815, doi:10.1056/NEJMe0810213 (2009).
- 15 Martínez-Reyes, I. & Chandel, N. S. Cancer metabolism: looking forward.

- Nature Reviews Cancer* **21**, 669–680, doi:10.1038/s41568-021-00378-6 (2021).
- 16 Martin-Perez, M., Urdiroz-Urricelqui, U., Bigas, C. & Benitah, S. A. The role of lipids in cancer progression and metastasis. *Cell Metab* **34**, 1675–1699, doi:10.1016/j.cmet.2022.09.023 (2022).
- 17 Platten, M., Nollen, E. A. A., Röhrig, U. F., Fallarino, F. & Opitz, C. A. Tryptophan metabolism as a common therapeutic target in cancer, neurodegeneration and beyond. *Nat Rev Drug Discov* **18**, 379–401, doi:10.1038/s41573-019-0016-5 (2019).
- 18 Lim, S. A., Su, W., Chapman, N. M. & Chi, H. Lipid metabolism in T cell signaling and function. *Nat Chem Biol* **18**, 470–481, doi:10.1038/s41589-022-01017-3 (2022).
- 19 Reina-Campos, M., Scharping, N. E. & Goldrath, A. W. CD8(+) T cell metabolism in infection and cancer. *Nat Rev Immunol* **21**, 718–738, doi:10.1038/s41577-021-00537-8 (2021).
- 20 Neinast, M., Murashige, D. & Arany, Z. Branched Chain Amino Acids. *Annu Rev Physiol* **81**, 139–164, doi:10.1146/annurev-physiol-020518-114455 (2019).
- 21 Ritterhoff, J. & Tian, R. Metabolic mechanisms in physiological and pathological cardiac hypertrophy: new paradigms and challenges. *Nat Rev Cardiol*, doi:10.1038/s41569-023-00887-x (2023).
- 22 Nie, M. *et al.* Evolutionary metabolic landscape from preneoplasia to invasive lung adenocarcinoma. *Nat Commun* **12**, 6479, doi:10.1038/s41467-021-26685-y (2021).
- 23 Shanmuganathan, M. *et al.* A Cross-Platform Metabolomics Comparison Identifies Serum Metabolite Signatures of Liver Fibrosis Progression in Chronic Hepatitis C Patients. *Front Mol Biosci* **8**, 676349, doi:10.3389/fmolb.2021.676349 (2021).
- 24 Xu, Z. *et al.* Efficient plasma metabolic fingerprinting as a novel tool for diagnosis and prognosis of gastric cancer: a large-scale, multicentre study. *Gut*, doi:10.1136/gutjnl-2023-330045 (2023).
- 25 Ren, S. *et al.* Integration of Metabolomics and Transcriptomics Reveals Major Metabolic Pathways and Potential Biomarker Involved in Prostate Cancer. *Mol Cell Proteomics* **15**, 154–163, doi:10.1074/mcp.M115.052381 (2016).
- 26 Mayerle, J. *et al.* Metabolic biomarker signature to differentiate pancreatic ductal adenocarcinoma from chronic pancreatitis. *Gut* **67**, 128–137, doi:10.1136/gutjnl-2016-312432 (2018).
- 27 Wang, P. P. *et al.* Serum Metabolomic Profiling Reveals Biomarkers for Early Detection and Prognosis of Esophageal Squamous Cell Carcinoma. *Front Oncol* **12**, 790933, doi:10.3389/fonc.2022.790933 (2022).
- 28 Matsumoto, T. *et al.* Targeted Metabolomic Profiling of Plasma Samples in Gastric Cancer by Liquid Chromatography–Mass Spectrometry. *Digestion* **104**, 97–108, doi:10.1159/000526864 (2023).
- 29 Kim, Y. L. *et al.* Metabolic alterations of short-chain fatty acids and TCA cycle intermediates in human plasma from patients with gastric cancer. *Life*

- Sci* **309**, 121010, doi:10.1016/j.lfs.2022.121010 (2022).
- 30 Guo, S., Wang, Y., Zhou, D. & Li, Z. Electric Field-Assisted Matrix Coating Method Enhances the Detection of Small Molecule Metabolites for Mass Spectrometry Imaging. *Anal Chem* **87**, 5860–5865, doi:10.1021/ac504761t (2015).
- 31 Nannini, G., Meoni, G., Amedei, A. & Tenori, L. Metabolomics profile in gastrointestinal cancers: Update and future perspectives. *World J Gastroenterol* **26**, 2514–2532, doi:10.3748/wjg.v26.i20.2514 (2020).
- 32 Bijlsma, S. *et al.* Large-scale human metabolomics studies: a strategy for data (pre-) processing and validation. *Anal Chem* **78**, 567–574, doi:10.1021/ac051495j (2006).
- 33 Nakada, Y. *et al.* Hypoxia induces heart regeneration in adult mice. *Nature* **541**, 222–227, doi:10.1038/nature20173 (2017).
- 34 Agathocleous, M. *et al.* Ascorbate regulates haematopoietic stem cell function and leukaemogenesis. *Nature* **549**, 476–481, doi:10.1038/nature23876 (2017).
- 35 Piskounova, E. *et al.* Oxidative stress inhibits distant metastasis by human melanoma cells. *Nature* **527**, 186–191, doi:10.1038/nature15726 (2015).
- 36 Kim, J. *et al.* CPS1 maintains pyrimidine pools and DNA synthesis in KRAS/LKB1-mutant lung cancer cells. *Nature* **546**, 168–172, doi:10.1038/nature22359 (2017).
- 37 Huang, F. *et al.* Inosine Monophosphate Dehydrogenase Dependence in a Subset of Small Cell Lung Cancers. *Cell Metab* **28**, 369–382. e365, doi:10.1016/j.cmet.2018.06.005 (2018).
- 38 Li, X. K. *et al.* Arginine deficiency is involved in thrombocytopenia and immunosuppression in severe fever with thrombocytopenia syndrome. *Sci Transl Med* **10**, doi:10.1126/scitranslmed.aat4162 (2018).
- 39 Xiao, N. *et al.* Integrated cytokine and metabolite analysis reveals immunometabolic reprogramming in COVID-19 patients with therapeutic implications. *Nat Commun* **12**, 1618, doi:10.1038/s41467-021-21907-9 (2021).
- 40 Pang, H. *et al.* Aberrant NAD(+) metabolism underlies Zika virus-induced microcephaly. *Nat Metab* **3**, 1109–1124, doi:10.1038/s42255-021-00437-0 (2021).
- 41 Zhao, J. *et al.* Metabolic remodelling during early mouse embryo development. *Nat Metab* **3**, 1372–1384, doi:10.1038/s42255-021-00464-x (2021).
- 42 Ishwaran, H., Kogalur, U. B., Blackstone, E. H. & Lauer, M. S. Random survival forests. *The Annals of Applied Statistics* **2**, 841–860, 820 (2008).
- 43 Bounsiar, A. & Madden, M. G. in *2014 International Conference on Information Science & Applications (ICISA)*. 1–4.
- 44 Agarwal, P. *et al.* Integrative analysis reveals novel associations between DNA methylation and the serum metabolome of adolescents with type 2 diabetes: A cross-sectional study. *Front Endocrinol (Lausanne)* **13**, 934706, doi:10.3389/fendo.2022.934706 (2022).
- 45 Hoel, F. *et al.* A map of metabolic phenotypes in patients with myalgic

- encephalomyelitis/chronic fatigue syndrome. *JCI Insight* **6**, doi:10.1172/jci.insight.149217 (2021).
- 46 Markin, P. A. *et al.* Plasma metabolomic profile in prostatic intraepithelial neoplasia and prostate cancer and associations with the prostate-specific antigen and the Gleason score. *Metabolomics* **16**, 74, doi:10.1007/s11306-020-01694-y (2020).
- 47 Tong, D. *et al.* MiR-22, regulated by MeCP2, suppresses gastric cancer cell proliferation by inducing a deficiency in endogenous S-adenosylmethionine. *Oncogenesis* **9**, 99, doi:10.1038/s41389-020-00281-z (2020).
- 48 Wang, Z. *et al.* NNMT enriches for AQP5(+) cancer stem cells to drive malignant progression in early gastric cardia adenocarcinoma. *Gut*, doi:10.1136/gutjnl-2022-328408 (2023).
- 49 Maldonado, L. Y., Arsene, D., Mato, J. M. & Lu, S. C. Methionine adenosyltransferases in cancers: Mechanisms of dysregulation and implications for therapy. *Exp Biol Med (Maywood)* **243**, 107–117, doi:10.1177/1535370217740860 (2018).
- 50 Dai, M. *et al.* Analysis of low-molecular-weight metabolites in stomach cancer cells by a simplified and inexpensive GC/MS metabolomics method. *Anal Bioanal Chem* **412**, 2981–2991, doi:10.1007/s00216-020-02543-6 (2020).
- 51 Khan, I. *et al.* LC/MS-Based Polar Metabolite Profiling Identified Unique Biomarker Signatures for Cervical Cancer and Cervical Intraepithelial Neoplasia Using Global and Targeted Metabolomics. *Cancers (Basel)* **11**, doi:10.3390/cancers11040511 (2019).
- 52 Klupczynska, A. *et al.* Determination of low-molecular-weight organic acids in non-small cell lung cancer with a new liquid chromatography–tandem mass spectrometry method. *J Pharm Biomed Anal* **129**, 299–309, doi:10.1016/j.jpba.2016.07.028 (2016).
- 53 Park, J., Shin, Y., Kim, T. H., Kim, D. H. & Lee, A. Plasma metabolites as possible biomarkers for diagnosis of breast cancer. *PLoS One* **14**, e0225129, doi:10.1371/journal.pone.0225129 (2019).
- 54 Chen, Y., Hu, L., Lin, H., Yu, H. & You, J. Serum metabolomic profiling for patients with adenocarcinoma of the esophagogastric junction. *Metabolomics* **18**, 26, doi:10.1007/s11306-022-01883-x (2022).
- 55 Lanser, L. *et al.* Inflammation-Induced Tryptophan Breakdown is Related With Anemia, Fatigue, and Depression in Cancer. *Front Immunol* **11**, 249, doi:10.3389/fimmu.2020.00249 (2020).
- 56 Kutluana, U., Kilciler, A. G., Mizrak, S. & Dilli, U. Can neopterin be a useful immune biomarker for differentiating gastric intestinal metaplasia and gastric atrophy from non-atrophic non-metaplastic chronic gastritis? *Gastroenterol Hepatol* **42**, 289–295, doi:10.1016/j.gastrohep.2019.01.005 (2019).
- 57 Wang, Y. P. *et al.* Effect of Helicobacter Pylori on Plasma Metabolic Phenotype in Patients With Gastric Cancer. *Cancer Control* **28**, 10732748211041881, doi:10.1177/10732748211041881 (2021).

- 58 Yildirim, H., Kavgaci, G., Chalabiyev, E. & Dizdar, O. Advances in the Early Detection of Hepatobiliary Cancers. *Cancers (Basel)* **15**, doi:10.3390/cancers15153880 (2023).
- 59 Gao, X. *et al.* Dietary methionine influences therapy in mouse cancer models and alters human metabolism. *Nature* **572**, 397–401, doi:10.1038/s41586-019-1437-3 (2019).
- 60 Loftfield, E. *et al.* Prospective Investigation of Serum Metabolites, Coffee Drinking, Liver Cancer Incidence, and Liver Disease Mortality. *J Natl Cancer Inst* **112**, 286–294, doi:10.1093/jnci/djz122 (2020).
- 61 Tao, J. *et al.* Targeting gut microbiota with dietary components on cancer: Effects and potential mechanisms of action. *Crit Rev Food Sci Nutr* **60**, 1025–1037, doi:10.1080/10408398.2018.1555789 (2020).
- 62 Meng, C., Bai, C., Brown, T. D., Hood, L. E. & Tian, Q. Human Gut Microbiota and Gastrointestinal Cancer. *Genomics Proteomics Bioinformatics* **16**, 33–49, doi:10.1016/j.gpb.2017.06.002 (2018).
- 63 Moustafa, T. *et al.* Alterations in lipid metabolism mediate inflammation, fibrosis, and proliferation in a mouse model of chronic cholestatic liver injury. *Gastroenterology* **142**, 140–151.e112, doi:10.1053/j.gastro.2011.09.051 (2012).
- 64 Li, J. *et al.* The Mediterranean diet, plasma metabolome, and cardiovascular disease risk. *Eur Heart J* **41**, 2645–2656, doi:10.1093/eurheartj/ehaa209 (2020).
- 65 Lam, B. Q., Srivastava, R., Morvant, J., Shankar, S. & Srivastava, R. K. Association of Diabetes Mellitus and Alcohol Abuse with Cancer: Molecular Mechanisms and Clinical Significance. *Cells* **10**, doi:10.3390/cells10113077 (2021).
- 66 Agopian, J. *et al.* GlcNAc is a mast-cell chromatin-remodeling oncometabolite that promotes systemic mastocytosis aggressiveness. *Blood* **138**, 1590–1602, doi:10.1182/blood.2020008948 (2021).
- 67 Gao, R. *et al.* Integrated Analysis of Colorectal Cancer Reveals Cross-Cohort Gut Microbial Signatures and Associated Serum Metabolites. *Gastroenterology* **163**, 1024–1037.e1029, doi:10.1053/j.gastro.2022.06.069 (2022).
- 68 Song, J. W. *et al.* Omics-Driven Systems Interrogation of Metabolic Dysregulation in COVID-19 Pathogenesis. *Cell Metab* **32**, 188–202.e185, doi:10.1016/j.cmet.2020.06.016 (2020).
- 69 Wang, H. *et al.* Serum metabolic traits reveal therapeutic toxicities and responses of neoadjuvant chemoradiotherapy in patients with rectal cancer. *Nat Commun* **13**, 7802, doi:10.1038/s41467-022-35511-y (2022).
- 70 van den Berg, R. A., Hoefsloot, H. C., Westerhuis, J. A., Smilde, A. K. & van der Werf, M. J. Centering, scaling, and transformations: improving the biological information content of metabolomics data. *BMC Genomics* **7**, 142, doi:10.1186/1471-2164-7-142 (2006).
- 71 Roy, S. *et al.* Diagnostic efficacy of circular RNAs as noninvasive, liquid

- biopsy biomarkers for early detection of gastric cancer. *Mol Cancer* **21**, 42, doi:10.1186/s12943-022-01527-7 (2022).
- 72 Sammut, S. J. *et al.* Multi-omic machine learning predictor of breast cancer therapy response. *Nature* **601**, 623–629, doi:10.1038/s41586-021-04278-5 (2022).
- 73 Niu, L. *et al.* Noninvasive proteomic biomarkers for alcohol-related liver disease. *Nat Med* **28**, 1277–1287, doi:10.1038/s41591-022-01850-y (2022).
- 74 Oh, T. G. *et al.* A Universal Gut-Microbiome-Derived Signature Predicts Cirrhosis. *Cell Metab* **32**, 878–888. e876, doi:10.1016/j.cmet.2020.06.005 (2020).
- 75 Huang, S. *et al.* Identification and Validation of Plasma Metabolomic Signatures in Precancerous Gastric Lesions That Progress to Cancer. *JAMA Netw Open* **4**, e2114186, doi:10.1001/jamanetworkopen.2021.14186 (2021).
- 76 Ikeda, A. *et al.* Serum metabolomics as a novel diagnostic approach for gastrointestinal cancer. *Biomed Chromatogr* **26**, 548–558, doi:10.1002/bmc.1671 (2012).
- 77 Yu, J., Zhao, J., Yang, T., Feng, R. & Liu, L. Metabolomics Reveals Novel Serum Metabolic Signatures in Gastric Cancer by a Mass Spectrometry Platform. *J Proteome Res* **22**, 706–717, doi:10.1021/acs.jproteome.2c00295 (2023).
- 78 Stefan-van Staden, R. I., Ilie-Mihai, R. M., Magerusan, L., Coros, M. & Pruneanu, S. Enantioanalysis of glutamine—a key factor in establishing the metabolomics process in gastric cancer. *Anal Bioanal Chem* **412**, 3199–3207, doi:10.1007/s00216-020-02575-y (2020).
- 79 Huang, R. *et al.* Metabolic Profiling of Urinary Chiral Amino-Containing Biomarkers for Gastric Cancer Using a Sensitive Chiral Chlorine-Labeled Probe by HPLC-MS/MS. *J Proteome Res* **20**, 3952–3962, doi:10.1021/acs.jproteome.1c00267 (2021).
- 80 Qu, Q., Zeng, F., Liu, X., Wang, Q. J. & Deng, F. Fatty acid oxidation and carnitine palmitoyltransferase I: emerging therapeutic targets in cancer. *Cell Death Dis* **7**, e2226, doi:10.1038/cddis.2016.132 (2016).
- 81 Tan, Y. *et al.* Adipocytes fuel gastric cancer omental metastasis via PITPNC1-mediated fatty acid metabolic reprogramming. *Theranostics* **8**, 5452–5468, doi:10.7150/thno.28219 (2018).
- 82 Chen, T., Wu, G., Hu, H. & Wu, C. Enhanced fatty acid oxidation mediated by CPT1C promotes gastric cancer progression. *J Gastrointest Oncol* **11**, 695–707, doi:10.21037/jgo-20-157 (2020).
- 83 Sucher, R. *et al.* Neopterin, a prognostic marker in human malignancies. *Cancer Lett* **287**, 13–22, doi:10.1016/j.canlet.2009.05.008 (2010).
- 84 Li, X. *et al.* Single-cell RNA sequencing reveals a pro-invasive cancer-associated fibroblast subgroup associated with poor clinical outcomes in patients with gastric cancer. *Theranostics* **12**, 620–638, doi:10.7150/thno.60540 (2022).
- 85 Isci Bostanci, E. *et al.* A New Diagnostic and Prognostic Marker in

- Endometrial Cancer: Neopterin. *Int J Gynecol Cancer* **27**, 754–758, doi:10.1097/igc.0000000000000952 (2017).
- 86 Pichler, R. *et al.* Predictive and prognostic role of serum neopterin and tryptophan breakdown in prostate cancer. *Cancer Sci* **108**, 663–670, doi:10.1111/cas.13171 (2017).
- 87 Nechita, V. I. *et al.* Chitotriosidase and Neopterin as Two Novel Potential Biomarkers for Advanced Stage and Survival Prediction in Gastric Cancer—A Pilot Study. *Diagnostics (Basel)* **13**, doi:10.3390/diagnostics13071362 (2023).
- 88 Ciocan, A. *et al.* Exploratory Evaluation of Neopterin and Chitotriosidase as Potential Circulating Biomarkers for Colorectal Cancer. *Biomedicines* **11**, doi:10.3390/biomedicines11030894 (2023).
- 89 Sekine, H. & Motohashi, H. Roles of CNC Transcription Factors NRF1 and NRF2 in Cancer. *Cancers (Basel)* **13**, doi:10.3390/cancers13030541 (2021).
- 90 Guo, Q. *et al.* ADMA mediates gastric cancer cell migration and invasion via Wnt/ β -catenin signaling pathway. *Clin Transl Oncol* **23**, 325–334, doi:10.1007/s12094-020-02422-7 (2021).
- 91 Smyth, E. C., Nilsson, M., Grabsch, H. I., van Grieken, N. C. T. & Lordick, F. Gastric cancer. *The Lancet* **396**, 635–648, doi:10.1016/s0140-6736(20)31288-5 (2020).
- 92 Tripathi, A. *et al.* Chemically informed analyses of metabolomics mass spectrometry data with Qemistree. *Nat Chem Biol* **17**, 146–151, doi:10.1038/s41589-020-00677-3 (2021).
- 93 Bauermeister, A., Mannocho-Russo, H., Costa-Lotuf, L. V., Jarmusch, A. K. & Dorrestein, P. C. Mass spectrometry-based metabolomics in microbiome investigations. *Nat Rev Microbiol* **20**, 143–160, doi:10.1038/s41579-021-00621-9 (2022).
- 94 Chakouri, N. *et al.* Fibroblast growth factor homologous factors serve as a molecular rheostat in tuning arrhythmogenic cardiac late sodium current. *Nat Cardiovasc Res* **1**, 1–13, doi:10.1038/s44161-022-00060-6 (2022).
- 95 Lu, W., Wang, L., Chen, L., Hui, S. & Rabinowitz, J. D. Extraction and Quantitation of Nicotinamide Adenine Dinucleotide Redox Cofactors. *Antioxid Redox Signal* **28**, 167–179, doi:10.1089/ars.2017.7014 (2018).
- 96 Li, L. *et al.* Characterization of Metabolic Patterns in Mouse Oocytes during Meiotic Maturation. *Mol Cell* **80**, 525–540. e529, doi:10.1016/j.molcel.2020.09.022 (2020).
- 97 Park, J. S. *et al.* Mechanical regulation of glycolysis via cytoskeleton architecture. *Nature* **578**, 621–626, doi:10.1038/s41586-020-1998-1 (2020).
- 98 Hakimi, A. A. *et al.* An Integrated Metabolic Atlas of Clear Cell Renal Cell Carcinoma. *Cancer Cell* **29**, 104–116, doi:10.1016/j.ccell.2015.12.004 (2016).
- 99 Pang, Z. *et al.* MetaboAnalyst 5.0: narrowing the gap between raw spectra and functional insights. *Nucleic Acids Res* **49**, W388–w396, doi:10.1093/nar/gkab382 (2021).
- 100 Berg, J. A. *et al.* Metaboverse enables automated discovery and

- visualization of diverse metabolic regulatory patterns. *Nat Cell Biol* **25**, 616–625, doi:10.1038/s41556-023-01117-9 (2023).
- 101 Denko, N. C. Hypoxia, HIF1 and glucose metabolism in the solid tumour. *Nat Rev Cancer* **8**, 705–713, doi:10.1038/nrc2468 (2008).
- 102 Mason, E. F. & Hornick, J. L. Succinate dehydrogenase deficiency is associated with decreased 5-hydroxymethylcytosine production in gastrointestinal stromal tumors: implications for mechanisms of tumorigenesis. *Mod Pathol* **26**, 1492–1497, doi:10.1038/modpathol.2013.86 (2013).
- 103 Xiao, M. *et al.* Inhibition of α -KG-dependent histone and DNA demethylases by fumarate and succinate that are accumulated in mutations of FH and SDH tumor suppressors. *Genes Dev* **26**, 1326–1338, doi:10.1101/gad.191056.112 (2012).
- 104 Killian, J. K. *et al.* Succinate dehydrogenase mutation underlies global epigenomic divergence in gastrointestinal stromal tumor. *Cancer Discov* **3**, 648–657, doi:10.1158/2159-8290.Cd-13-0092 (2013).
- 105 Hui, S. *et al.* Glucose feeds the TCA cycle via circulating lactate. *Nature* **551**, 115–118, doi:10.1038/nature24057 (2017).
- 106 Sun, C. *et al.* Spatially resolved multi-omics highlights cell-specific metabolic remodeling and interactions in gastric cancer. *Nat Commun* **14**, 2692, doi:10.1038/s41467-023-38360-5 (2023).
- 107 Niu, Q. *et al.* Expression of 5-HT Relates to Stem Cell Marker LGR5 in Patients with Gastritis and Gastric Cancer. *Dig Dis Sci* **68**, 1864–1872, doi:10.1007/s10620-022-07772-6 (2023).
- 108 Field, M. S., Lan, X., Stover, D. M. & Stover, P. J. Dietary Uridine Decreases Tumorigenesis in the Apc(Min/+) Model of Intestinal Cancer. *Curr Dev Nutr* **2**, nzy013, doi:10.1093/cdn/nzy013 (2018).
- 109 Li, X. *et al.* Metabolomics based plasma biomarkers for diagnosis of oral squamous cell carcinoma and oral erosive lichen planus. *J Cancer* **13**, 76–87, doi:10.7150/jca.59777 (2022).
- 110 Li, M. *et al.* Discovery and Validation of Potential Serum Biomarkers with Pro-Inflammatory and DNA Damage Activities in Ulcerative Colitis: A Comprehensive Untargeted Metabolomic Study. *Metabolites* **12**, doi:10.3390/metabol12100997 (2022).
- 111 Wang, W. *et al.* Identification of the γ -glutamyl cycle as a novel therapeutic target and 5-oxoproline as a new biomarker for diagnosing pancreatic cancer. *Ann Med* **55**, 2242247, doi:10.1080/07853890.2023.2242247 (2023).
- 112 Chen, X. *et al.* Characterization of 5-oxo-L-prolinase in normal and tumor tissues of humans and rats: a potential new target for biochemical modulation of glutathione. *Clin Cancer Res* **4**, 131–138 (1998).
- 113 Joshi, S. S. & Badgwell, B. D. Current treatment and recent progress in gastric cancer. *CA Cancer J Clin* **71**, 264–279, doi:10.3322/caac.21657 (2021).

- 114 Sasako, M. *et al.* Five-year outcomes of a randomized phase III trial comparing adjuvant chemotherapy with S-1 versus surgery alone in stage II or III gastric cancer. *J Clin Oncol* **29**, 4387-4393, doi:10.1200/jco.2011.36.5908 (2011).
- 115 Fong, C., Johnston, E. & Starling, N. Neoadjuvant and Adjuvant Therapy Approaches to Gastric Cancer. *Curr Treat Options Oncol* **23**, 1247-1268, doi:10.1007/s11864-022-01004-9 (2022).
- 116 Jiang, L., Lee, S. C. & Ng, T. C. Pharmacometabonomics Analysis Reveals Serum Formate and Acetate Potentially Associated with Varying Response to Gemcitabine-Carboplatin Chemotherapy in Metastatic Breast Cancer Patients. *J Proteome Res* **17**, 1248-1257, doi:10.1021/acs.jproteome.7b00859 (2018).

Reviewers' Comments:

Reviewer #1:

Remarks to the Author:

Some of my concerns and issues are resolved by the authors and they provided adequate explanation/data to support their conclusion. However, there are a few remaining concerns that should be addressed before accepting the manuscript.

A discussion about future applicability of the proposed ML models is missing. Here are some important issues that should be emphasized:

- As previously stated, using normalized peak area, normalized based on QC samples and scaled later would limit the use of these ML models for future suspect patients. In other words, in order for the ML to understand the GC prognosis or its risk with respect to the new samples, these samples need to be analyzed together and in the presence of QC samples that are used in ML model structure while training the models.
- No limits or threshold values are provided on the levels of important metabolites to understand normal range versus problematic ranges, as it is done on the other conventional diagnostic tools. This means that data treatment of new patients should be done by experts or online (codes or app are not provided by authors) after observing the author's workflow (LC-MS and ML modelling workflow).
- There are no uncertainty measurements or application domain provided by authors. A typical ML model should be accompanied by uncertainty assessment to provide the chances that ML will fail to correctly discriminate between different risks or diagnosis results of GC (between 0-100). For instance, if a new case is completely outlier or out of ranges known to ML, what will be the outcome of ML models?

Page 6, line 118: rephrase "differentially abundant" to statistically different?

Page 6, lines 115-126: Although I understand the author's intention about preparing and publishing the analytical method elsewhere, it is vital to provide ME and Rec data for these metabolites in each sample to ensure that the variation is sample relevant and it is not caused by instrumental or analytical method shortcomings. A discussion about matrix effect is highly encouraged to give confidence about the use of peak area instead of concentration data.

Page 7-8, all discussion about 10 DM model and 28 PM model: looking back to Figure 2, GSSG looks significantly down regulated in contrast to Uridine, Succinate etc, while it is not included in both models. What is the reason for this? Could be the use of ratio GSH/GSSG which confuses the LASSO in 28PM model and 10DM model?

Page 6, lines 115-126: I do appreciate for providing LOD and LOQ values.

Page 12, lines 321-322: but the authors have not used quantitative metabolomics to achieve this conclusion here.

Page 12, lines 324: Apart from LOQ and LOD, authors did not provide any other means in the SIF or the main text to assure about "quantification accuracy". Actually, they used only normalized peak area which is still acceptable for generic metabolomics, but there are few limitations about it, as listed above.

Page 21, lines 664-665: I do understand the author's intention about exclusion of these samples, but what could be interesting to see is how ML model would react to these treated samples which are completely new to it. This is particularly of interest, since the authors did not provide applicability domain (a method to define boundary of a ML model with the respect to training set data range and using it to show if ML will make predictions on the basis of learned information from previously trained cases) of their tool.

Page 21, line 677, "EDTA-blood samples", define the abbreviation.

Page 22, lines 684-686: what was the final reconstituted volume?

Page 22, lines 690-711: I do appreciate providing LOQ and LOD in SIF for the detected metabolites. After reading this paper (Nature Communications, (2023) 14:7172) and the extraction method from the authors, there are few issues noticed. Are these LOQ and LOD data, instrumental or method data (mLOD and mLOQ)? To measure a LOD of 1pM, there should be significant level of pre-concentration to be adopted during sample preparation on real samples, right? This could increase ME substantially. Please, explicitly comment on these data, how they are obtained.

Why did not the authors provide quantitative results if target analysis is performed? Authors have the reference standards of many of the metabolites detected (SIF, "428955_1_supp_8150661_s23z3n.xlsx") or those which are part of ML models and it is definitely beneficial for ease of future applications to use concentration data.

Page 31, Figure 3: Authors have performed extra validation by switching cases between test set and training set and created models to test whether the ML models are sensitive to individual cases in the training set or not. Test set 2 seems to have somewhat lower accuracy than Test set 1 with many NGC grouped with Stage IA and IB. It could be beneficial to study this effect further by permutation test and double cross-validation analysis (it is applicable to other ML methods as well: Metabolomics 8 (Suppl 1), 3-16 (2012)) to verify that there is no such risk.

Reviewer #2:

Remarks to the Author:

The authors have sufficiently addressed my concerns, and the inclusion of the comparative analysis with 3-biomarker panel and the 10-DM model is appreciated and strengthens the generalizability of the work

Reviewer #3:

Remarks to the Author:

The authors have thoroughly engaged with my questions and suggestions. I note that everything has been satisfactorily addressed.

Dear Reviewers,

We sincerely appreciate the editors and reviewers for your positive, constructive, and insightful comments, which significantly help us to improve our manuscript. In the revised version of the manuscript, we provided substantial revisions, including:

- 1) We conducted an assessment of matrix effects and recovery rates using representative isotopic standards of metabolites from different classes in both GC and NGC samples. The corresponding data have been included in **Supplementary Data 3**.
- 2) In response to the reviewer's query regarding the quantification accuracy of our mass spectrometry tools, we supplied pertinent methodological data to affirm the reliability of our relative quantitative results. Additionally, we conducted an absolute quantitative analysis using the isotope internal standard, showcasing the method's potential for accurate absolute quantification (**Supplementary Data 3**). Moreover, we executed an absolute quantitative analysis on a small cohort, and the corresponding data is presented in **Supplementary Data 4**.
- 3) To enhance transparency and reproducibility, we have uploaded the codes used for model construction and corresponding "Read Me" files to our GitHub repository (https://github.com/Yangzi-Chen2023/GC_NC-Res). Notably, we have specifically emphasized information regarding uncertainty measurements within the code, addressing the reviewer's concern related to the Machine Learning (ML) model.
- 4) We have incorporated a discussion section addressing the limitations of the model's applications and providing an outlook on future plans for applying machine learning models in a clinical context.

In addition, we have made extensive efforts to provide a point-by-point response (highlighted in blue) to each point raised by the editors and reviewers. We hope that these responses have addressed all the concerns/advice, and improved our manuscript substantially.

Please note that figures and tables exclusively presented in this point-by-point response letter are labeled as "Response Letter Figure/Table," while others are referenced by the same serial numbers as those in the revised manuscript.

Once again, we would like to express our sincere gratitude for your time and efforts. Thank you very much!

Point-by-point responses to reviewers' comments

Reviewer #1 (Remarks to the Author):

Some of my concerns and issues are resolved by the authors and they provided adequate explanation/data to support their conclusion. However, there are a few remaining concerns that should be addressed before accepting the manuscript.

A discussion about future applicability of the proposed ML models is missing. Here are some important issues that should be emphasized:

- As previously stated, using normalized peak area, normalized based on QC samples and scaled later would limit the use of these ML models for future suspect patients. In other words, in order for the ML to understand the GC prognosis or its risk with respect to the new samples, these samples need to be analyzed together and in the presence of QC samples that are used in ML model structure while training the models.
- No limits or threshold values are provided on the levels of important metabolites to understand normal range versus problematic ranges, as it is done on the other conventional diagnostic tools. This means that data treatment of new patients should be done by experts or online (codes or app are not provided by authors) after observing the author's workflow (LC-MS and ML modelling workflow).
- There are no uncertainty measurements or application domain provided by authors. A typical ML model should be accompanied by uncertainty assessment to provide the chances that ML will fail to correctly discriminate between different risks or diagnosis results of GC (between 0-100). For instance, if a new case is completely outlier or out of ranges known to ML, what will be the outcome of ML models?

Response: We appreciate your time and efforts in reviewing our manuscript, along with your positive and constructive comments aimed at strengthening our paper.

- As pointed out, our current models exhibit limitations in practical applications. The current models rely on normalized peak area for relative quantification, necessitating concurrent sample testing and determination in the presence of QC samples. We have incorporated pertinent clarifications in the Discussion section on lines 408-423 of pages 14-15: "It's essential to understand that current models are not yet appropriate for direct application in clinical settings due to the following limitation. Given that the model is constructed based on relatively quantitative metabolics data, understanding the GC prognosis or risk regarding to new patients using these machine learning (ML) models necessitates the concurrent presence of quality control (QC) samples employed during the ML model construction process. However, our

study identified crucial metabolites capable of distinguishing between GC and NGC, representing a significant step towards constructing a model with potential clinical applications. In the future, to further advance the translation of our research into clinical practice, we intend to conduct absolute quantitative metabolomics with large-scale multicenter patient samples using isotopic internal standards of these key metabolites. This will help to elucidate the normal range and problematic range of the important metabolites, thereby determining detection thresholds for GC. Additionally, we will explore alternative detection methods for differential metabolites, including simplified mass spectrometry detection methods and novel detection strategies such as assay kits, aiming to streamline detection time and costs and facilitate clinical applications.”

- However, it should be noted that our study involves a broad-spectrum screening of 258 metabolites, with the principal objective of identifying metabolites capable of distinguishing between GC and NGC, followed by the construction of models. To fulfill this objective, the widely acknowledged approach of relatively quantitative targeted metabolomics is employed¹⁻⁶. Looking ahead, to further advance the translational aspects of our research into clinical practice, we plan to conduct large-scale, multicenter absolute quantitative metabolomic assays employing isotopic internal standards for model metabolites. This endeavor aims to determine both the normal range and problematic ranges of model metabolites, thereby establishing the detection threshold for GC. Furthermore, we will explore alternative detection methods for model metabolites, potentially developing detection kits, and implementing strategies to streamline detection time and cost, thereby facilitating clinical application.
- As Reviewer suggested that we have provided the code for feature selection and model construction on **GitHub** (https://github.com/Yangzi-Chen2023/GC_NC-Res). As illustrated in Fig. 3d, the model yields a predicted value (between 0-1) for each new case, quantifying the model's uncertainty in prediction. For instance, a predicted value of 0.86 indicates that the model is 86% confident in predicting the given case as GC. Details on the model's predicted values for all samples is available in **Supplementary Data 5**. Furthermore, as mentioned in the Data Analysis and Preprocessing section of the Methods, we generated scaled data (between 0-1) by comparing the normalized peak area of each metabolite to the sum of the normalized peak area from all detected metabolites in the given sample. Consequently, new cases share the same data range as the training data, and instances exceeding the model's detection range are generally uncommon.

1. Page 6, line 118: rephrase “differentially abundant” to statistically different?

Response: We thank for the reviewer for the suggestion and have already rephrased the text as suggested on line 121 of page 6.

2. Page 6, lines 115-126: Although I understand the author’s intention about preparing and publishing the analytical method elsewhere, it is vital to provide ME and Rec data for these metabolites in each sample to ensure that the variation is sample relevant and it is not caused by instrumental or analytical method shortcomings. A discussion about matrix effect is highly encouraged to give confidence about the use of peak area instead of concentration data.

Response: We are grateful for the reviewer's suggestion to include matrix effect and recovery data for the metabolites in various samples, thereby ensuring the reliability of our data. We conducted these experiments and supplemented the relevant data in the modified supplementary materials, as shown in **Supplementary Data 3**.

While conducting matrix effects and recovery rate experiments, we thoroughly adhere to the **ICH (International Council for Harmonization of Technical Requirements for Pharmaceuticals for Human Use) guidelines**, surpassing the evaluation criteria of metabolomics methodologies. According to the ICH guideline (M10), the matrix effect should be evaluated by analyzing at least 3 replicates of low and high QCs, each prepared using a matrix from at least 6 different sources/lots. Here, we evaluated the matrix effect by analyzing 3 replicates of low and high QCs, each prepared using a matrix from 28 different sources including the plasma from 14 GC patients and 14 non-GC controls (randomly selected). A similar design was adopted for the assessment of recovery rates.

Unlike exogenous drugs, it is crucial to emphasize that all 258 metabolites in our method are endogenous chemicals. Due to the absence of a suitable blank plasma matrix for directly assessing their matrix effects and recovery rates, we have opted to use stable isotope-labeled metabolites as substitutes for endogenous metabolites to obtain pertinent methodological data. Considering that the 258 metabolites covered by our method belong to distinct classes and display diverse physicochemical properties, we diligently endeavored to obtain representative isotopic standards from different classes for conducting relevant experiments. Eventually, we selected seven isotope-labeled metabolites, including arginine (amino acids), methionine (amino acids), choline (cholines), adenosine ribose (nucleosides), nicotinamide (vitamins), glucose (saccharides) and succinate (organic acids) to perform relevant experiment, and the

precision data of matrix factor and recovery rates (peak area) were shown in the Response Letter Table 1.

As shown in Supplementary Data 3, it can be observed that the presence of plasma matrix indeed leads to signal suppression of the metabolites (matrix factor < 1, peak area vs. peak area) due to matrix effect. However, the precisions (percent coefficient of variation (%CV)) of the matrix factors obtained from 28 different sources are all within the threshold of 15% (refer to Response Letter Table 1), meeting the requirements specified in the ICH guidelines. **Therefore, it is verified that the variations between GC patients and non-GC controls are not from the matrix difference.** At the same time, we assessed the recovery rates to illustrate the impact of sample extraction. **It can be verified that the extraction step does not appear to significantly impact the relative quantification of metabolites, as indicated by Response Letter Table 2.** Therefore, it can be concluded that the variations between GC patients and non-GC controls are sample-relevant and are not caused by instrumental or analytical method shortcomings.

Response Letter Table 1. Matrix effect evaluation of seven isotope-labeled metabolites

Analyte	RSD of matrix factor (GC, n=14)			RSD of matrix factor (non-GC, n=14)			RSD of matrix factor (RSD, n=28)		
	Low	Medium	High	Low	Medium	High	Low	Medium	High
Arginine- ¹³ C6	7.50%	5.72%	6.88%	5.50%	5.65%	5.83%	6.51%	5.79%	6.27%
Methionine- ¹³ C5	2.89%	3.85%	4.52%	3.19%	3.40%	4.06%	3.34%	3.70%	4.28%
Choline-trimethyl-D9	6.01%	6.53%	4.14%	3.54%	5.84%	4.32%	5.24%	7.84%	4.15%
Adenosine-ribose- ¹³ C5	5.84%	6.70%	4.82%	3.09%	3.18%	1.73%	4.58%	5.16%	3.57%
Nicotinamide-D4	2.66%	3.79%	4.77%	2.10%	3.15%	3.27%	2.53%	3.59%	4.16%
Glucose- ¹³ C6	13.94%	5.58%	9.39%	6.20%	3.84%	6.75%	10.92%	4.71%	8.05%
Succinate- ¹³ C2	6.42%	6.59%	14.10%	3.24%	4.61%	6.55%	5.26%	6.77%	10.83%

Response Letter Table 2. Recovery evaluation of seven isotope-labeled metabolites

Analyte	RSD of recovery rate (GC, n=14)			RSD of recovery rate (non-GC, n=14)			RSD of recovery rate (RSD, n=28)		
	Low	Medium	High	Low	Medium	High	Low	Medium	High
Arginine- ¹³ C6	6.77%	4.40%	5.58%	5.98%	7.82%	7.32%	6.28%	6.28%	6.42%

Methionine- ¹³ C5	4.30%	4.01%	5.21%	3.87%	3.86%	4.84%	4.04%	3.86%	5.26%
Choline-trimethyl-D9	11.49%	13.22%	4.29%	8.02%	7.41%	6.64%	10.33%	11.29%	5.90%
Adenosine-ribose- ¹³ C5	5.25%	4.05%	3.84%	3.42%	5.08%	3.90%	4.46%	5.01%	4.96%
Nicotinamide-D4	4.21%	3.60%	3.44%	3.22%	3.96%	5.70%	3.90%	4.09%	5.06%
Glucose- ¹³ C6	8.43%	6.28%	4.88%	6.80%	4.68%	5.47%	7.64%	5.44%	5.64%
Succinate- ¹³ C2	4.29%	5.93%	11.00%	8.95%	4.29%	7.40%	6.93%	5.90%	9.34%

3. Page 7-8, all discussion about 10 DM model and 28 PM model: looking back to Figure 2, GSSG looks significantly down regulated in contrast to Uridine, Succinate etc, while it is not included in both models. What is the reason for this? Could be the use of ratio GSH/GSSG which confuses the LASSO in 28PM model and 10DM model?

Response: We appreciate your insights. **This could be attributed to the distinct advantages and preferences of Lasso and differential metabolic analysis in feature selection⁷.** Although GSSG exhibits remarkable significance in differential metabolic analysis, its exclusion in the Lasso feature selection could be attributed to high correlations with other features.

Differential metabolic analysis is typically employed to identify features significantly associated with differences between groups, focusing solely on the average differences of features between groups without considering inter-feature correlations. In contrast, the primary objective of Lasso feature selection is to reduce model complexity and enhance generalizability by selecting features correlated with the target variable. Consequently, Lasso may overlook features with weak correlations to the target variable but significant differences between groups.

It is essential to elucidate that the GSH/GSSG ratio has not been employed in either of the models. Additionally, we explored the inclusion of the GSH/GSSG ratio in the feature selection process. Regardless of whether the GSH/GSSG ratio was considered, neither GSSG nor the GSH/GSSG ratio were incorporated into the model.

4. Page 6, lines 115-126: I do appreciate for providing LOD and LOQ values.

Response: Thanks for your response. To further substantiate the reliability of our detection method and enhance confidence, we have included matrix effect, recovery data, and code information in **Supplementary Data 3** and GitHub.

5. Page 12, lines 321-322: but the authors have not used quantitative metabolomics to achieve this conclusion here.

Response: Thanks for pointing out this issue. We have already rephrased the sentences on lines 324 and 327 of page 12.

6. Page 12, lines 324: Apart from LOQ and LOD, authors did not provide any other means in the SIF or the main text to assure about “quantification accuracy”. Actually, they used only normalized peak area which is still acceptable for generic metabolomics, but there are few limitations about it, as listed above.

Response: We acknowledge the reviewer's concern regarding the limitation of relative quantification and a supplementary discussion was provided on this limitation in the revised manuscript on lines 408-423 of pages 14-15.

However, relative quantification methods are widely utilized in metabolomics research to identify variations in metabolite composition under different conditions, such as differences between disease and normal states, changes at different time points, and so on^{2,4-6}. As stated in the response to question 2, It has been verified that the matrix effect and extraction step will not significantly impact the relative quantification results of metabolites using our method. Meanwhile, using our analytical method for relative quantification analysis on different samples demonstrated excellent intra-day and inter-day precision, as shown in **Supplementary Data 1**. In addition, during the plasma metabolomic analysis of GC patients and non-GC controls, we randomized and blinded the samples to avoid the impact of instrument fluctuations and employed QC samples to remove potential inter-batch variations. **These procedures ensure a high level of confidence in the relative quantitative analysis results obtained in our study.**

Furthermore, to demonstrate the potential of our analytical method for accurate quantification, we performed absolute quantification analysis using this method on seven representative isotopic standards. As shown in **Response Letter Table 3**, the relative standard deviations (RSD) for the seven stable isotope-labeled metabolites were all below 15% and their relative errors (RE) were within the range of $\pm 10\%$, surpassing the requirements outlined in the ICH guidelines (RSD < 15% and RE were with the range of 15%).

Response Letter Table 3. Precision and accuracy for the determination of seven isotope-labeled metabolites in human plasma (n = 8).

Analyte	Nominal	Mean±SD	Precision	Accuracy
	Con. (uM)	uM	RSD (%)	RE (%)
Arginine- ¹³ C6	2	1.975±0.016	0.804	-1.255
	5	4.927±0.026	0.526	-1.460
	10	10.078±0.082	0.810	0.776
Methionine- ¹³ C5	0.2	0.198±0.001	0.481	-1.052
	2	1.915±0.030	1.580	-4.271
	10	9.685±0.163	1.681	-3.150
Choline-trimethyl-D9	0.01	0.010±0.001	11.185	2.996
	0.5	0.497±0.003	0.633	-0.522
	2	2.024±0.022	1.085	1.194
Adenosine-ribose- ¹³ C5	0.015	0.015±0.001	4.156	-2.152
	0.15	0.145±0.003	1.942	-3.200
	0.75	0.761±0.011	1.459	1.486
Nicotinamide-D4	0.20	0.192±0.003	1.515	-3.893
	2	1.904±0.046	2.400	-4.797
	10	9.548±0.229	2.400	-4.520
Glucose- ¹³ C6	1000	974.934±25.447	2.610	-2.507
	5000	4995.196±129.383	2.590	-0.096
	10000	9186.056±317.834	3.460	-8.139
Succinate- ¹³ C2	1	0.997±0.037	3.727	-0.271
	5	5.029±0.066	1.303	0.578
	20	19.580±0.328	1.676	-2.099

7. Page 21, lines 664-665: I do understand the author's intention about exclusion of these samples, but what could be interesting to see is how ML model would react to these treated samples which are completely new to it. This is particularly of interest, since the authors did not provide applicability domain (a method to define boundary of a ML model with the respect to training set data range and using it to show if ML will make predictions on the basis of learned information from previously trained cases) of their tool.

Response: We appreciate the questions raised by the reviewers. **We posit that the plasma of patients undergoing treatment may have been influenced by drug interventions or therapeutic measures, leading to changes in their metabolic features and biases in the predicted outcomes.** Consequently, the results derived from the computation of model predictions relying on these perturbed metabolic features cannot serve as an effective assessment of the accuracy of the model predictions.

Even though, in response to your inquiry, we utilized plasma samples from patients who have completed neoadjuvant therapy to conduct requisite model tests. However, our findings reveal that when employing the 10-DM model for classification, the rate of accurate predictions for these neoadjuvant patients was notably low in comparison to untreated GC or NGC cases with an accuracy of 0.435 (**Response Letter Fig. 1a**). Furthermore, as elucidated in the preceding response, the application of the 28-PM model to neoadjuvant patients yielded a predicted C-INDEX value of 0.5 (**Response Letter Fig. 1b**). This indicates that the model's ability to discriminate between high and low risk in neoadjuvant patients is comparable to random chance (with a probability of correctness at 0.5). This result underscores the model's unsuitability for prognostic risk assessment in this patient cohort.

Considering the challenges posed by dimensionality and feature weights, we refrained from determining a threshold based on the distribution of training set data or distance metrics to assess whether a sample falls within the model's applicability domain⁸. **Instead, we embraced an ensemble learning approach to discern potential model uncertainties and boundary cases.** Specifically, we employed the bootstrap method, performing multiple random samplings with replacement on the training data to train several independent models. The final prediction was determined through a voting mechanism, where the model yielded a predicted value ranging from 0 to 1 for each new case. We established a threshold of 0.5, representing a majority consensus among the independent models, as the diagnostic criterion for the model. If the predicted value for a case exceeds 0.5, the model diagnoses it as belonging to the class labeled GC. For reference, we have uploaded the code, functional annotations, and model output results employed by the model to GitHub as supplementary materials.

Response Letter Fig. 1 | a, Diagnostic prediction was performed utilizing the 10-DM model for gastric cancer patients who underwent neoadjuvant chemotherapy (NC-GC, represented by yellow circles), untreated gastric cancer patients (GC, represented by purple circles), and non-gastric cancer controls (NGC, represented by green circles). The dashed line, delineated at the threshold value of 0.5, segregates the participants

into the predicted NGC group and predicted GC group. **b**, Prognostic prediction of the GC patients who received neoadjuvant chemotherapy using the 28-PM model. The dotted line, drawn at the cut-off value of 2.1, divides the patients into high- and low-risk groups. Green circles and gray circles represent survived and deceased respectively.

8. Page 21, line 677, “EDTA-blood samples”, define the abbreviation.

Response: Thanks for the question from the reviewer. We have rephrased the sentences to “blood was drawn using BD Vacutainer EDTA tubes” on lines 675-676 of page 21.

9. Page 22, lines 684-686: what was the final reconstituted volume?

Response: The reconstituted volume is 50µl and the final injection volume is 10µl as described on lines 690 and 698 of page 22.

10. Page 22, lines 690-711: I do appreciate providing LOQ and LOD in SIF for the detected metabolites. After reading this paper (Nature Communications, (2023) 14:7172) and the extraction method from the authors, there are few issues noticed. Are these LOQ and LOD data, instrumental or method data (mLOD and mLOQ)? To measure a LOD of 1pM, there should be significant level of pre-concentration to be adopted during sample preparation on real samples, right? This could increase ME substantially. Please, explicitly comment on these data, how they are obtained.

Why did not the authors provide quantitative results if target analysis is performed? Authors have the reference standards of many of the metabolites detected (SIF, “428955_1_supp_8150661_s23z3n.xlsx”) or those which are part of ML models and it is definitely beneficial for ease of future applications to use concentration data.

Response: We would like to express our gratitude for the reviewer's comments regarding the LOD and LOQ data. Upon reviewing the referenced paper supplied by the reviewer, it's crucial for us to clarify that **the LOD and LOQ data we presented were determined solely based on the methodology itself**, independent of any calculations generated by the algorithm provided by the instrument manufacturer.

We take the Response Letter Fig. 2 as an example to illustrate how we obtained the data. As shown in **Response Letter Fig. 2**, the chromatograms of three representative metabolites in the blank control (representing solvent noise), as well as

samples at LOD and LOQ concentrations, are provided. The LOD and LOQ for the metabolites were determined following established analytical guidelines, specifically defined as LOD (S/N > 3, calculated by the peak height) and LOQ (S/N > 10, calculated by the peak height). While the related S/N calculated by the algorithm provided by the instrument manufacturer is much higher.

Response Letter Fig. 2 | Chromatograms of three representative metabolites at LOD and LOQ concentrations

It is imperative to emphasize that the determined Limit of Detection (LOD) and Limit of Quantification (LOQ) data are exclusively derived from mixed standard solutions of metabolite standards in aqueous solutions. Given that all 258 metabolites covered by our methodology are water-soluble, negligible matrix discrepancies were observed during the preparation process of methodological investigation samples at varying concentrations. It is important to note that when

analyzing real samples (blood, cells, tissues), the associated data may be subject to variations due to matrix effects.

It is crucial to emphasize that targeted metabolomic analysis does not equate to absolute quantitative analysis. Relative quantitative analysis methods are frequently employed in targeted metabolomics, especially in large-scale targeted analyses of metabolites for specific biological contexts^{2,4-6}. Compared to untargeted metabolomics, targeted metabolomic offers several advantages. First, enhanced qualitative precision is achieved in targeted metabolomics by simultaneous monitoring of the precursor ions, product ions, and retention times of metabolites during analysis. Second, targeted metabolomics employs tandem mass spectrometry for quantification, providing heightened sensitivity, an expanded linear range, and reduced risk of detector saturation. In the end, unlike conventional untargeted metabolomics, which focuses solely on differentiating metabolite features, targeted metabolomics allows for relative quantification assessment of both changing and unchanged metabolites within crucial metabolic pathways.

Performing broad-spectrum absolute quantification of metabolites in plasma poses significant challenges. Unlike exogenous drugs, the absence of a suitable blank plasma matrix makes it difficult to construct an accurate calibration curve for the quantification of endogenous metabolite. Therefore, isotope-labeled metabolites are commonly selected as substitutes for constructing standard curves, enabling precise quantification of metabolites. However, due to the diverse physicochemical properties exhibited by the metabolites covered in the method, the use of one or a few isotopic internal standards may not accurately simulate the actual behavior of metabolites during LC-MS analysis. However, the introduction of an excessive number of isotopic internal standard channels in targeted metabolomics (MRM) analysis may adversely impact the accuracy of quantitative results.

Despite the challenges mentioned earlier, in accordance with the reviewers' request, we selected six isotopic internal standards for concentration determination. Utilizing the principle of structural similarity, we performed absolute quantification analysis on the metabolites of interest within a small study cohort ($n_{GC} = 36$, $n_{non-GC} = 36$), and the relative results was shown in Supplementary Data 4. It should be noted that the selection of isotopic internal standards for absolute quantification can impact the accuracy of quantitative results. Due to limitations in the size of the sample set, the associated results are to be considered indicative or for reference purposes only. In our future research, we intend to utilize isotopic internal standards specific to the screened compounds for their absolute quantification across a large, multicenter sample set. This approach aims to construct early screening and prognostic models

that are more conducive to clinical applications.

11. Page 31, Figure 3: Authors have performed extra validation by switching cases between test set and training set and created models to test whether the ML models are sensitive to individual cases in the training set or not. Test set 2 seems to have somewhat lower accuracy than Test set 1 with many NGC grouped with Stage IA and IB. It could be beneficial to study this effect further by permutation test and double cross-validation analysis (it is applicable to other ML methods as well: Metabolomics 8 (Suppl 1), 3–16 (2012)) to verify that there is no such risk.

Response: We feel great thanks for your professional review work on our article. Indeed, in contrast to permutation tests and double cross-validation⁹, we employed a bootstrap method which is more suitable for random forest model to prevent overfitting¹⁰. It is essential to elucidate that Test set 1 and Test set 2 were not involved in the construction and optimization processes of the model. Particularly, Test set 2 represents an external dataset sourced from a different center compared to the Training set and Test set 1. As illustrated in Fig. 3a, during the model training process, the random forest model employed the bootstrap method to repetitively sample the original data, conducting 100 iterations to estimate parameter distributions. In contrast to Cross-Validation and permutation tests, Bootstrap mitigates sample reduction issues by resampling, making it particularly suitable for statistically inferring population distribution characteristics in small sample scenarios.

Additionally, we optimized model hyperparameters through grid search, including the number of decision trees (n_estimators), maximum tree depth (max_depth), and minimum sample split (min_sample_split). The optimal model performance was achieved when n_estimators=100, max_depth=None, and min_sample_split=2.

Evidently, we observed that Test Set 2 exhibited lower accuracy compared to Test Set 1. We speculate that this difference stems from the diverse origins of Test Set 2, introducing variations due to factors such as collection methods and environmental influences. In future work, we aim to collect data from multiple centers to mitigate such risks and enhance the robustness of our findings.

Fig. 3 | Machine learning-derived prediction model based on plasma metabolome for GC diagnosis.

a, Design of the modeling workflow. LASSO regression and random forest algorithm were adopted for feature selection and model training. The 10-DM model was validated in a test set and an external test set. **b**, The Receiver operating characteristic (ROC) curve for the diagnosis of GC patients in the test set. A 95% confidence interval was calculated based on the mean and covariance of one thousand random sampling tests. **c**, Contribution of the ten metabolites to the 10-DM model. **d-g**, The prediction performance of the 10-DM model for distinguish GC from NGC in the test set 1 (**d**) and the test set 2 (**e**) and for distinguishing stage I GC patients from NGC in the test set 1 (**f**) and the test set 2 (**g**). The dotted line represented the cut-off value of 0.50 used to separate the predicted NGC (on the left side) from GC (on the right side).

Reviewer #2 (Remarks to the Author):

The authors have sufficiently addressed my concerns, and the inclusion of the comparative analysis with 3-biomarker panel and the 10-DM model is appreciated and strengthens the generalizability of the work

Response: We highly appreciate your time and efforts in reviewing our manuscript and your insightful comments and suggestions for our work.

Reviewer #3 (Remarks to the Author):

The authors have thoroughly engaged with my questions and suggestions. I note that everything has been satisfactorily addressed.

Response: We appreciate your time and efforts in reviewing our manuscript and your positive and constructive comments to strengthen our paper.

Reference

- 1 Kachroo, P. *et al.* Metabolomic profiling reveals extensive adrenal suppression due to inhaled corticosteroid therapy in asthma. *Nat Med* **28**, 814-822, doi:10.1038/s41591-022-01714-5 (2022).
- 2 Chen, Y. *et al.* Development of a Data-Independent Targeted Metabolomics Method for Relative Quantification Using Liquid Chromatography Coupled with Tandem Mass Spectrometry. *Anal Chem* **89**, 6954-6962, doi:10.1021/acs.analchem.6b04727 (2017).
- 3 Luo, P. *et al.* A Large-scale, multicenter serum metabolite biomarker identification study for the early detection of hepatocellular carcinoma. *Hepatology* **67**, 662-675, doi:10.1002/hep.29561 (2018).
- 4 Cao, G. *et al.* Large-scale targeted metabolomics method for metabolite profiling of human samples. *Anal Chim Acta* **1125**, 144-151, doi:10.1016/j.aca.2020.05.053 (2020).
- 5 Li, L. *et al.* Characterization of Metabolic Patterns in Mouse Oocytes during Meiotic Maturation. *Mol Cell* **80**, 525-540.e529, doi:10.1016/j.molcel.2020.09.022 (2020).
- 6 Agathocleous, M. *et al.* Ascorbate regulates haematopoietic stem cell function and leukaemogenesis. *Nature* **549**, 476-481, doi:10.1038/nature23876 (2017).
- 7 Motamedi, F., Pérez-Sánchez, H., Mehridehnavi, A., Fassihi, A. & Ghasemi, F. Accelerating Big Data Analysis through LASSO-Random Forest Algorithm in QSAR Studies. *Bioinformatics* **38**, 469-475, doi:10.1093/bioinformatics/btab659 (2022).
- 8 Sutton, C. *et al.* Identifying domains of applicability of machine learning models for materials science. *Nat Commun* **11**, 4428, doi:10.1038/s41467-020-17112-9 (2020).
- 9 Szymańska, E., Saccenti, E., Smilde, A. K. & Westerhuis, J. A. Double-check: validation of diagnostic statistics for PLS-DA models in metabolomics studies. *Metabolomics* **8**, 3-16, doi:10.1007/s11306-011-0330-3 (2012).
- 10 Breiman, L. Random Forests. *Machine Learning* **45**, 5-32, doi:10.1023/A:1010933404324 (2001).

Reviewers' Comments:

Reviewer #1:

Remarks to the Author:

The authors have addressed my concerns.